# LANGUAGE MODEL AGENTS SUFFER FROM COMPOSITIONAL GENERALIZATION IN WEB AUTOMATION

## ABSTRACT

Language model agents (LMA) recently emerged as a promising paradigm on muti-step decision making tasks, often outperforming humans and other reinforcement learning agents. Despite the promise, their performance on real-world applications that often involve combinations of tasks is still underexplored. In this work, we introduce a new benchmark, called *CompWoB* – 50 new compositional web automation tasks reflecting more realistic assumptions. We show that while existing prompted LMAs (`gpt-3.5-turbo` or `gpt-4`) achieve 94.0% average success rate on base tasks, their performance degrades to 24.9% success rate on compositional tasks. On the other hand, transferred LMAs (finetuned only on base tasks) show less generalization gap, dropping from 85.4% to 54.8%. By balancing data distribution across tasks, we train a new model, HTML-T5++, that surpasses human-level performance (95.2%) on MiniWoB, and achieves the best zero-shot performance on CompWoB (61.5%). While these highlight the promise of small-scale finetuned and transferred models for compositional generalization, their performance further degrades under different instruction compositions changing combinational order. In contrast to the recent remarkable success of LMA, our benchmark and detailed analysis emphasize the necessity of building LMAs that are robust and generalizable to task compositionality for real-world deployment.

## 1 INTRODUCTION

Based on the exceptional capability of large language models (LLMs) (OpenAI, 2023; Anil et al., 2023; Touvron et al., 2023) in commonsense understanding (Brown et al., 2020; Chowdhery et al., 2022), multi-step reasoning (Wei et al., 2022; Kojima et al., 2022), program synthesis (Chen et al., 2021) and self-improvement (Shinn et al., 2023; Madaan et al., 2023; To et al., 2023), language model agents (LMA) have recently emerged to tackle various decision making problems, such as robotics (Huang et al., 2022a; Ahn et al., 2022), information retrieval (Nakano et al., 2021; Yao et al., 2022b), and external tool use (Wu et al., 2023; Shen et al., 2023; Lu et al., 2023). Especially, in web automation (Shi et al., 2017), LMAs with prompting (Kim et al., 2023; Sun et al., 2023; Zheng et al., 2023) outperform humans and other learning-based agents, such as reinforcement learning (Humphreys et al., 2022) or finetuned language models (Gur et al., 2022; Furuta et al., 2023).

Despite their proficiency in MiniWoB (Shi et al., 2017), a standard web automation benchmark, it is still unclear whether LMAs could deal with challenges in the real world: such as complex observation (Gur et al., 2023), domain generalization (Deng et al., 2023), and ambiguity of instructions (Zhou et al., 2023b). These challenges are exacerbated due to the open-ended nature of real-world tasks, making it infeasible to prepare exemplars and prompts in advance for any unseen task.

In this work, we extensively study the generalization of LMAs to more realistic task compositions. We first design a new controlled test bed, called CompWoB, with 50 compositional tasks by combining a set of base tasks based on their difficulty (Figure 1). Each compositional task is implemented from 2 to 8 base tasks in a single-page or multi-page environment with instructions linked together using simple connectors such as "and then". Only providing the knowledge about base tasks, we investigate the generalization performance of existing SoTA prompted LMAs (Kim et al., 2023; Sun et al., 2023; Zheng et al., 2023) with planning, self-improvement, program synthesis, and structured prompts that are supported by `gpt-3.5-turbo` and `gpt-4`. Our findings indicate that their performance drops significantly, from 94.0% success on base tasks to 24.9% success on compositional tasks. In contrast,

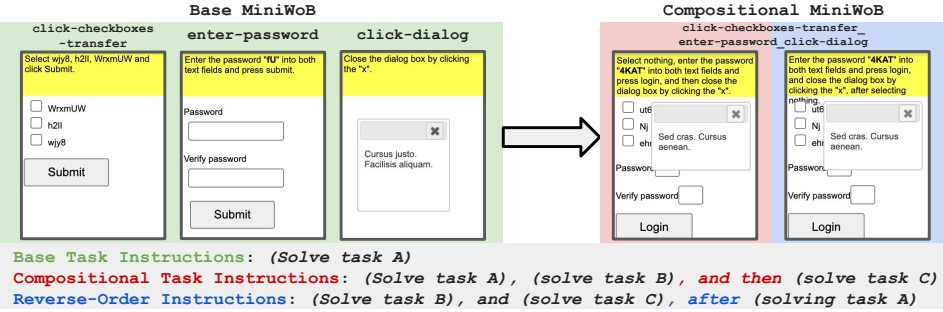

Figure 1: We design *CompWoB*, as a novel decision making benchmark for LMAs, by leveraging the high composability of simulated web environments. We first select base tasks from the original MiniWoB (Shi et al., 2017) based on the brute-force task complexity averaged among existing LMAs. While considering the feasibility, we randomly combine them into a single task (e.g. `click-checkboxes-transfer` + `enter-password` + `click-dialog`→ `click-checkboxes-transfer_enter-password_click-dialog`). The instructions of base tasks are stitched with "**and then**". LMAs are asked to satisfy the given instructions sequentially (e.g. satisfying the success criteria of task $A \to B \to C$). We also implement **reverse-order instruction** settings, where the instructions are provided upside down (e.g. solve task $B$, and solve task $C$, after solving task $A$). These complex yet controllable strategies make the analysis of LMA's behaviors easy, while maintaining complex and ambiguous aspects of real-world web environments.

small-scale LMAs finetuned only on base tasks and zero-shot-transferred to compositional settings (i.e. transferred LMAs), deal with unknown task compositionality better, achieving 54.8% success rate on average. By rebalancing the data distribution, we also train a new model, HTML-T5++, that achieves human-level performance on MiniWob and performs the best among all the LMAs on compositional tasks. We further show that LMAs struggle to handle complex instruction compositions permuting the order of sub-instructions, where prompted agents are more robust to the difference in the order of compositions compared to transferred agents (6.9% vs 23.8% drop in performance). Finally, we illustrate that instruction length and observation complexity are useful indicators of compositional task performance.

In contrast to the recent notable success of LMAs, our benchmark and detailed analysis highlight building robust and generalizable LMAs to be safely deployed in the real world. In summary, our key contributions are:

- We empirically show that (1) **prompted LMAs even with** `gpt-4` **suffer from generalizing to compositional web automation tasks much more than transferred LMAs**, and (2) **LMAs are highly sensitive to the order of instructions**.
- We develop *CompWoB*[1], simulated web environments for LMAs to measure the generalization to the realistic task compositionality and complex instructions.
- We propose a new data mixture strategy for finetuning LMA. HTML-T5++, trained on this strategy, achieves human-level performance on MiniWoB (95.2%) and the best zero-shot transfer to CompWoB (61.5%).

## 2 RELATED WORKS

**Language Model Agents** Beyond the common NLP tasks, LLMs could act as autonomous agents (Wang et al., 2023b; Qin et al., 2023a) to solve the given instruction-following tasks, by considering the context in the prompt as states (Ahn et al., 2022; Yao et al., 2022b; Hsieh et al., 2023) and sequentially planning and manipulating external "tools" or "actuators", such as calculators (Parisi et al., 2022), retrievers (Schick et al., 2023; Hao et al., 2023), APIs (Qin et al., 2023b; Tang et al., 2023a), programs (Gao et al., 2023; Wang et al., 2023a; Liang et al., 2023; Song et al., 2023; Cai et al., 2023), robotic commands (Huang et al., 2022a;b; Tang et al., 2023b), computer game (Nottingham et al., 2023; Wang et al., 2023c), or other foundation models (Lu et al., 2023; Hsieh et al., 2023; Wu et al., 2023; Shen et al., 2023; Yang et al., 2023). Those prior works have worked on proposing novel benchmarks (Li et al., 2023; Xu et al., 2023; Patil et al., 2023) and comparing backbone LLMs

---

[1]We will release the code in de-anonymized version.

(e.g. open-sourced v.s. private) (Ruan et al., 2023; Liu et al., 2023b;a). Despite their success, it is still unclear how such LMAs designed for specific tasks can generalize out-of-domain problems, which should be an important perspective since we may not prepare prompts and exemplars for all the possible combinations of problems in the real world. In this work, we measure the compositional generalization and robustness to the complex instructions in web automation.

**Web Automation** Although prior works have worked on imitation learning and reinforcement learning (Liu et al., 2018; Gur et al., 2019; Jia et al., 2019; Humphreys et al., 2022), web automation has become a popular domain as a benchmark for LMAs (Gur et al., 2022; Kim et al., 2023). In earlier work, finetuned LMAs, based on at most 3-billion parameters, amortize the training costs with the strong prior knowledge on web environments (Gur et al., 2022; Furuta et al., 2023; Shaw et al., 2023), but they often result in sub-optimal performances due to the insufficient data coverage. Recently, by leveraging capable private LLMs (Brown et al., 2020; Ouyang et al., 2022) with self-refinement (Kim et al., 2023), program synthesis (Sun et al., 2023), well-designed structured prompts with instruction-state translation (Zheng et al., 2023), or hierarchical prompts (Sridhar et al., 2023; Ma et al., 2023), prompted LMAs with few-shot exemplars have outperformed finetuned LMAs and shown competitive performance to humans and RL-finetuned agents. In contrast, our work discusses the generalization and robustness of those LMAs in zero-shot web automation with a set of compositional tasks, and resolves the sub-optimality of finetuned LMAs via data-rebalancing.

In addition to MiniWoB (Shi et al., 2017), a representative web simulator, several works have conducted real-world evaluation (Gur et al., 2023) and proposed novel benchmarks reflecting real-world assumptions, such as a simulated e-commerce site (Yao et al., 2022a), sand-boxed real-world websites (Zhou et al., 2023b), an adaptive sim-to-real bridge with unsupervised auto-curricula (Gur et al., 2021), and large-scale web interaction dataset curated by human annotators (Deng et al., 2023). However, real-world web automation may make the analysis challenging because it often faces many obstacles, such as complex HTML observations, domain gaps between websites, and ambiguous instructions. In this work, we design CompWoB while controlling task difficulty and ambiguity of instructions, and investigate what factors may prevent the generalization capability of LMAs in compositional tasks.

**Large Language Models for Compositional Tasks** Several works have investigated compositional natural language problems with LLMs, such as semantic parsing (Furrer et al., 2021; Shaw et al., 2021; Zhou et al., 2023a), logic grid puzzles (Dziri et al., 2023), mathematical reasoning (Chen et al., 2023), programming (Zelikman et al., 2023), and planning (Brahman et al., 2023), which shows that dynamical selection of exemplars for decomposed sub-problems (Drozdov et al., 2023) or model scaling (Qiu et al., 2022) could help generalization. While those are focused on static tasks, our paper studies compositional generalization in decision making, especially, in web automation where the task may have more explicitly decomposable structures (Gur et al., 2021) than natural language tasks.

## 3 PRELIMINARIES

Web automation could be described as a deterministic sequential decision making problem, which consists of a state space $\mathcal{S}$, action space $\mathcal{A}$, deterministic transition function $T : \mathcal{S} \times \mathcal{A} \to \mathcal{S}$, a set of instructions $\mathcal{G}$, a set of contextual information (i.e. prompts for LLM) $\mathcal{C}$, and episodic reward function (i.e. success criteria) $r : \mathcal{S} \times \mathcal{G} \times \mathcal{A} \to \{0, 1\}$. At each time step $t$, the language model agent $\pi$ infers the action conditioned on the prompt, instruction, current state, and previous actions $\pi : \mathcal{S} \times \underbrace{\mathcal{A} \times \cdots \times \mathcal{A}}_{\times t} \times \mathcal{C} \times \mathcal{G} \to \mathcal{A}$, and moves to the next state: $s_{t+1} = T(s_t, a_t)$. When the agent reaches the terminal state (e.g. Login button is clicked) or the max time step is exceeded, the episode is marked as a success if the instruction $g$ is satisfied (i.e. $r(s_t, g, a_t) = 1$). The state $s_t \in \mathcal{S}$ is a raw HTML, and we assume the programmatic action space: function(selector, text). function is either *click*, *move* or *type*, selector is an integer index or XPath that can uniquely specify the element, and text is a text input for *type* function.

**Task Compositionality** Web automation tasks can be decomposed into a set of primitive base tasks. For instance, (1) clicking several checkboxes, (2) fulfilling the password form, and (3) closing the dialog window. Such a combination could be open-ended. In this work, we assume that the task $\psi \in \Psi$ is characterized by a corresponding subtree of HTML ($\mathcal{S}_\psi \subset \mathcal{S}$) and instructions ($\mathcal{G}_\psi \subset \mathcal{G}$), and can be combined each other as long as the task is feasible and executable.

# 4 LANGUAGE MODEL AGENTS FOR WEB AUTOMATION

We here review the existing LMAs for web automation problems, such as RCI (Section 4.1), Ada-Planner (Section 4.2), Synapse (Section 4.3), and transferred LMAs (Section 4.4). To clarify their algorithmic difference, we further provide the pseudo code for prompted LMAs in Appendix B. Furthermore, we resolve the sub-optimal performance of transferred LMAs by proposing data-rebalanced finetuning (Section 4.5).

## 4.1 RCI

The agent with Recursive Criticism and Improvement (RCI) prompting (Kim et al., 2023) first generates an open-loop plan to follow a given instruction using few-shot demonstrations. Next, it uses a prompt-based critic to identify the problems in the plan and improves the plan by reflecting on self-criticized outputs, which is referred as an explicit RCI (ERCI) loop. After ERCI, the agent follows the self-improved plan step-by-step. Before executing the action at each time step, the agent grounds the action to the current state (i.e. HTML, open-loop plan, and previous actions) and refines its formatting to be parsable, which increases the feasibility and reduces hallucinations. These final steps are referred to as an implicit RCI (IRCI) loop without the self-criticism.

All of those play complementary roles to achieve proficient performance. While 1 ERCI and 3 IRCI loops are recommended by Kim et al. (2023), we observe that the optimal number of self-improvement iterations may differ across the tasks. See Appendix C.1 for the details.

## 4.2 ADAPLANNER

In contrast to other prompted LMAs, AdaPlanner (Sun et al., 2023) leverages the capability of program synthesis in LLMs to mitigate the hallucination in a plan. Conditioning on the instruction, the description of permissible actions in the web environments, and few-shot demonstrations, the agent first generates an open-loop plan in a Python function, where each snippet corresponds to the action. Once the agent receives environmental feedback at each step, such as assertion errors in the code, other functional errors, or "ask" action to LLMs, it adaptively re-generates the plan for the remaining steps in a closed-loop manner. It has been reported that LLMs more capable of code generation perform better, such as `text-davinci-003` than `gpt-3.5-turbo`.

## 4.3 SYNAPSE

Synapse (Zheng et al., 2023) argues that LMAs perform better if well-designed structured prompts are provided, even without self-improvement or program synthesis. The structured prompting is formed by two pre-processing strategies: state filtering and task reformulation. State filtering gradually transforms raw HTML into simple formatted text, such as a Pythonic list or dictionary, in a multi-step manner, which may improve the state understanding of LMAs. Task reformulation translates given instructions or raw HTML into decomposed queries: for instance, translating *"select 12/03/2016 as the date and hit submit"* into *"select the datepicker at step 1, click 'Prev' 7 times at step 2-8 (May is 7 months before December), click the date '12' at step 9, and finally submit at step 10"* (translated instruction), or mapping proper noun into corresponding XPath (translated HTML).

While detailed structured prompts have led to strong performances, those should be specialized for each primitive task in MiniWoB by leveraging 7 different types of reformulation. See Appendix C.3 for further details.

| Models | Dataset Size | Success Rate |
|---|---|---|
| HTML-T5 | 347K episodes | 85.6% |
| **HTML-T5++** (ours) | 424K (+77K) | 94.1% |
| | 361K (+77K - 63K) | 94.6% |
| | 332K (+77K - 92K) | 94.8% |
| | **292K (+77K - 132K)** | **95.2%** |
| | 259K (+77K - 165K) | 95.0% |
| RCI (Kim et al., 2023) | | 90.6% |
| AdaPlanner (Sun et al., 2023) | | 92.9% |
| Human | | 93.5% |
| CC-Net (Humphreys et al., 2022) | | 93.5% |
| RCI (`gpt-4`) (Kim et al., 2023) | | 94.0% |
| Synapse (Zheng et al., 2023) | | **98.5%** |

Table 1: Average success rate of finetuned LMAs in 56 tasks on MiniWoB. Adding 77K episodes and reducing redundant thousands of episodes, HTML-T5++ achieves competitive performance to prompted LMAs, RL-finetuned agents, and humans, while improving the success rate from 85.6% to 95.2%.

## 4.4 FINETUNED AND TRANSFERRED LANGUAGE MODEL AGENTS

In addition to the prompted LMAs, LMAs finetuned on base tasks have also been developed (Gur et al., 2022; Furuta et al., 2023; Shaw et al., 2023), which are built on pre-trained language models, such as T5 (Raffel et al., 2020), Flan-T5 (Chung et al., 2022), HTML-T5 (Gur et al., 2023), or Pix2Struct (Lee et al., 2023), with web automation demonstrations. Those LMAs take HTML (or screenshots) and previous actions as inputs and predict the text-format next actions in a closed-loop manner. Since pre-trained language models have a sufficient inductive bias for web environments and instruction-following, finetuned LMAs can data-efficiently achieve competitive performance to the RL-finetuned agents trained from scratch with domain-specific architectures (Humphreys et al., 2022; Liu et al., 2018). Compared to the prompted LMAs relying on private LLM API, it is possible to build on-premise agents based on tractable-size models (at most 3 billion parameters), which may reduce inference time and costs. However, prior works have pointed out that finetuned LMAs struggle to the sub-optimal performance (Zheng et al., 2023) and they require demonstrations on the order of hundreds of thousands while prompted LMAs just need hundreds of episodes (Kim et al., 2023). In this paper, we extensively evaluate such finetuned LMAs in **zero-shot transfer** settings; LMAs are finetuned only with base task demonstrations and should deal with unseen compositional tasks. We call those *transferred* LMAs in the later sections.

## 4.5 DATA-REBALANCING IMPROVES FINETUNED LANGUAGE MODEL AGENTS

Furuta et al. (2023) utilized agent-driven data collection instead of humans to improve the performance of finetuned LMAs further. For each task, 10k demonstrations are collected and filtered based on task success, which resulted in challenging tasks having much less than 10k demonstrations due to the sub-optimal performance of LMA on these tasks. We identify that by fixing the data-imbalance problem, the performance of finetuned LMAs can be significantly improved, achieving super-human performance on MiniWoB. We first run Synapse (Zheng et al., 2023) on MiniWoB and collect 77K additional demonstrations across 16 tasks on top of 347K demonstrations (Furuta et al., 2023) to compensate for the lack of data in specific tasks. We then estimate the "brute-force" task difficulty averaging success rates for representative web automation agents. Based on those proximal measures, we classify 65 base tasks into three categories, such as `easy` (0.8 - 1.0), `medium` (0.6 - 0.8), and `hard` (0.0 - 0.6) (see Appendix D). We then balance the number of episodes based on the task difficulty, where we gradually reduce the ratio of easier tasks to focus more on challenging tasks. For instance, we remove $X\%$ episodes from top-$k$ tasks in `easy` group (see Appendix E for the details).

We finetune HTML-T5-XL (Gur et al., 2023), a pre-trained language model with local and global attention in the encoder and a mixture of long-span denoising, on these rebalanced datasets. Table 1 shows that all the data-rebalance strategies improve the success rate, and reducing 50% episodes from `easy` tasks (finally 292K episodes in total) is the most effective rebalancing strategy. This suggests that finetuned LMAs can be as capable as prompted LMAs in decision making tasks. We include HTML-T5++ as a baseline in the following sections.

## 5 DESIGNING COMPWOB TO MEASURE COMPOSITIONAL GENERALIZATION

Even though MiniWoB includes a spectrum of simulated environments, they have still focused on narrow and single-task instances. We need more advanced environments to measure the generalization to the various challenges in real-world web automation, such as complex observation (Deng et al., 2023), instruction (Zhou et al., 2023b), and task compositionality (Gur et al., 2021). We design *CompWoB* (i.e. compositional MiniWoB), as a novel test bed for LMAs, by leveraging the high composability of simulated web environments (Figure 1). In CompWoB, we systematically combine several base tasks (from 2 to 8) in the original MiniWoB, which LMAs can already solve, into a single task (e.g. `click-checkboxes-transfer` + `enter-password` + `click-dialog` → `click-checkboxes-transfer_enter-password_click-dialog`). This allows us to control the complexity of HTML and instructions, ensuring novel tasks are solvable to some extent. CompWoB includes realistic task compositions, such as combining form filling, popup message, and login page, in a single or across multiple pages.

As mentioned in Section 4.5, we first calculate the average success rates among representative web automation agents as brute-force task complexity, and classify 65 primitive tasks in MiniWoB into 3

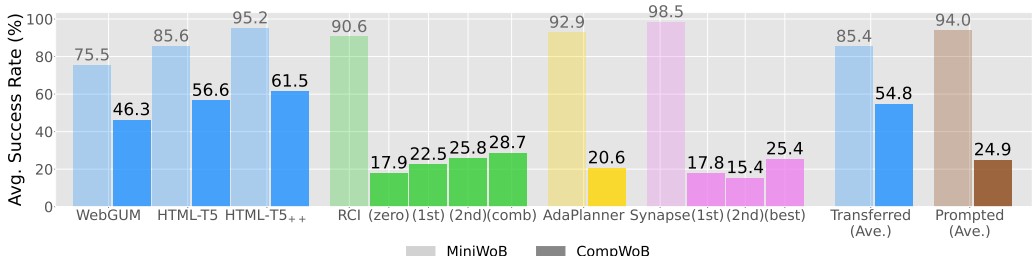

Figure 2: Average success rate of LMAs in 50 CompWoB tasks. The light color represents the performance in the original MiniWoB, and the dark color for CompWoB. We use `gpt-3.5-turbo` as the backbone LLM for prompted LMAs (RCI (Kim et al., 2023), AdaPlanner (Sun et al., 2023), Synapse (Zheng et al., 2023)), and transferred LMAs with 3-billion parameters. Transferred LMA, especially HTML-T5++, achieves the best generalization in compositional tasks, suppressing the performance degradation (from 95.2% to 61.5%). On the contrary, prompted LMAs drop their performance significantly; even the best RCI that uses combined task prompts in the composition just achieves 28.7% success (from 90.6% in base tasks). This indicates, in contrast to the base task performances, prompted LMAs are more vulnerable to, and transferred LMAs can deal with unknown task compositionality better than expected.

categories (`easy`, `medium`, `hard`) based on those scores. The details of classification are described in Appendix D. We randomly select base tasks from `easy` group, filter those combinations by their feasibility, and make 50 compositional tasks. We divide those into five categories: **two-way** tasks (20), **three-way** tasks (10), **n-way** tasks (5), **transition** tasks (5), and **easy-medium two-way** tasks (10). In n-way tasks, we combine from 4 to 8 tasks sequentially, and in transition tasks, we implement explicit page transition, for instance, transiting from the login form to the email browser. We also sample several tasks from `medium` group to construct easy-medium two-way tasks. You can find the full list of tasks in Appendix H. We simply stitch the instructions with "*and then*" and put each HTML on the same depth. LMAs should satisfy the given instructions sequentially, such as from task *A*, *B*, to *C*. Moreover, to test whether LMAs can deal with complex and ambiguous instructions, we propose **reverse-order instruction** settings, where the instruction is provided upside down while its task order is semantically the same (e.g. solve task *B* and *C*, after solving *A*). These simple yet controllable strategies make the analysis of LMA's behaviors tractable while reflecting compositional aspects of real-world tasks.

## 6   RESULTS

**Evaluation Methodology**  We evaluate both transferred and prompted LMAs with base MiniWoB demonstrations on the unseen compositional tasks in a "**zero-shot**" manner; i.e. **we do not provide any demonstrations on the compositional tasks for the training corpus and exemplars to measure the generalization**. We test 50 compositional tasks and run 100 episodes per task. We adopt `gpt-3.5-turbo` as a backbone LLM, unless otherwise mentioned. We assume the optimal exemplar retriever throughout experiments and always provide the pre-defined prompts to LMAs. We borrow hyper-parameters and prompt templates from respective papers with minimal change to respect our zero-shot transfer setting.

**RCI**  We test 4 prompting strategies: (1) zero-shot (without any exemplars), few-shot with (2) first-task exemplars, (3) second-task exemplars, and (4) combination exemplars (i.e. both first and second tasks). For consistency and limited context length, we always consider the first two tasks even if the number of primitive tasks is more than two, and fix the number of self-improvement iterations to 1 explicit RCI and 3 implicit RCI as recommended in the original paper. The exemplars we use are provided by Kim et al. (2023).

**AdaPlanner**  Following the original implementation, we use the exemplars provided by Sun et al. (2023) for the tasks where those base tasks are included, such as `enter-text` and `click-widget` (see Appendix C.2). Otherwise, the agents are prompted in a zero-shot manner.

**Synapse**  We test 3 prompting strategies: few-shot with (1) first-task exemplars, (2) second-task exemplars, and (3) best exemplars (i.e. maximum score between (1) and (2)). Because prompts and

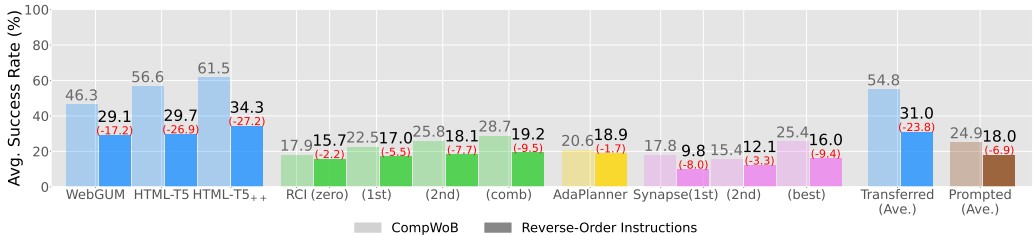

Figure 3: Average success rate of language model agents in **reverse-order instruction** settings. We use `gpt-3.5-turbo` as the backbone LLM for prompted LMAs (RCI, AdaPlanner, Synapse), and transferred LMAs with 3-billion parameters. Notably, most LMAs significantly degrade the success rate when reverse-order instructions are provided. This trend is more remarkable in transferred LMA (from 54.8% to 31.0% on average) than prompted LMA (from 24.9% to 18.0% on average), which suggests that any kind of LMAs are susceptible to the order of compositional instructions. The capability as general language models might be important to parse semantically complex instructions into the correct plan.

modules are quite different among the primitive tasks, we do not merge the prompts and just use proper hyper-parameters corresponding to the given exemplars designed by Zheng et al. (2023).

## 6.1 LANGUAGE MODEL AGENTS STRUGGLE TO HANDLE TASK COMPOSITIONALITY

Figure 2 shows that, in CompWoB, all the LMAs face performance degradation. Among those, transferred LMAs achieve better success rate (54.8%) than prompted LMAs (24.9%) on average. In particular, HTML-T5++ achieves the best generalization while suppressing the performance drop from 95.2% to 61.5%. In contrast, prompted LMAs degrade their performance drastically; even the best RCI with few-shot combination exemplars (comb) just degrades the success rate to 28.7% from 90.6% in base MiniWoB. These results indicate that LMAs suffer from generalization to task compositionality, and transferred LMAs can relatively deal with that better than prompted LMAs, which is an opposite trend to base MiniWoB performance. Among prompted LMAs, RCI performs better than AdaPlanner and Synapse, which suggests that multiple iterations of self-criticism and improvement might be more robust to out-of-domain decision making from the exemplars than program synthesis with feedback or structured prompting with state translations.

In the failure episodes (Table 2), LMAs often miss necessary steps, common to all the prompted LMAs. Since the instructions get long in compositional settings, LMAs may skip important intermediate steps to satisfy the instructions. In addition, they predict incorrect action types and XPath: for instance, hallucination in XPath (RCI) and mixing up *click* and *type* action (Synapse).

## 6.2 REVERSE-ORDER INSTRUCTIONS DEGRADE LANGUAGE MODEL AGENTS

As shown in Figure 3, all the LMAs significantly degrade the success rate when reverse-order instructions are provided. This trend is more remarkable in transferred LMAs dropping from 54.8% to 31.0% than prompted LMAs from 24.9% to 18.0% on average, which suggests that any kind of LMAs is susceptible to the order of compositional instructions and that transferred LMAs may not generalize well to diverse instructions beyond the dataset distribution. As opposed to Section 6.1, the performance differences among prompted LMAs are marginal, which implies that existing prompting methods, even with self-improvement, may not handle complex task instructions enough. The stronger capability as general-purpose language models or other prompting methods might be important to parse semantically complex instructions into the executable sequential plan.

| RCI (Kim et al., 2023) | | AdaPlanner (Sun et al., 2023) | | Synapse (Zheng et al., 2023) | |
|---|---|---|---|---|---|
| *Click button ONE, then click button TWO, and then select whX, 1Nk, fUK3 and click Submit* | | *Enter the password "UBKR" into both text fields, and then select KwpUv and click Submit* | | *Select yE, and then enter "Juan" into the text field and press Submit* | |
| ✔ | ✘ | ✔ | ✘ | ✔ | ✘ |
| 1. click //button[@id="subbtn1"] | 1. type //button[@id="subbtn2"] | 1. click //*[@id="password"] | | 1. click //*[text()="yE"]/input | 1. click //input[@id="tt"] |
| 2. click //button[@id="subbtn2"] | 2. click //*[text()="whX"]/input | 2. type UBKR | 1. type UBKR | 2. click //input[@id="tt"] | 2. type yE |
| 3. click //*[text()="whX"]/input | 3. click //*[text()="1Nk"]/input | 3. click //*[@id="verify"] | | | |
| 4. click //*[text()="1Nk"]/input | 4. click //*[text()="fUK3"]/input | 4. type UBKR | 2. type UBKR | 3. type Juan | 3. type Juan |
| 5. click //*[text()="fUK3"]/input | 5. click //*[text()="gSm"]/input | 5. click //input[@id="ch0"] | 3. click //input[@id="ch0"] | 4. click //*[@id="subbtn"] | 4. click //*[@id="subbtn"] |
| 6. click //*[@id="subbtn"] | 6. click //*[@id="subbtn"] | 6. click //*[@id="subbtn"] | 4. click //*[@id="subbtn"] | | |

Table 2: Failure examples in CompWoB. The left columns have correct plans and the right columns have failure plans. LMAs often ignore necessary intermediate steps or predict incorrect action types and XPath.

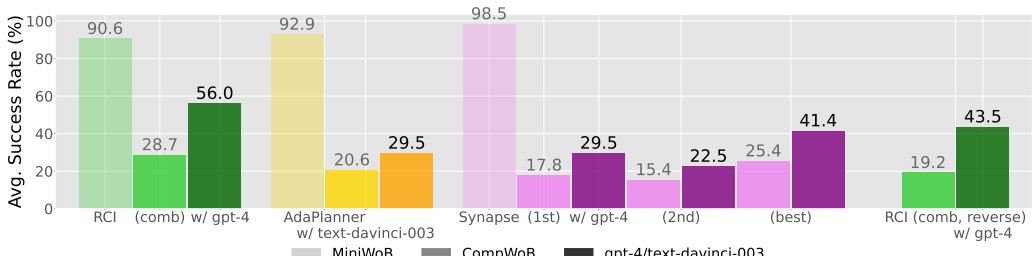

Figure 4: Average success rate of large language models with advanced LLMs (`gpt-4` for RCI and Synapse, and `text-davinci-003` for AdaPlanner). The lighter color represents the performance in MiniWoB, the medium color does in CompWoB with `gpt-3.5-turbo`, and the darker color does with `gpt-4` or `text-davinci-003`. The more capable models (`gpt-4` as a generalist and `text-davinci-003` as a coder) can improve the success rate of prompted LMAs but still struggle to generalize to compositional tasks (e.g. 56.0% by RCI) or to deal with reverse-order instructions (e.g. 43.5% by RCI). This may indicate that we need much better foundation models to realize deployable LMA in the complex real world.

Compared to Section 6.1, LMAs cannot parse reverse-order instructions into plans correctly (Table 3), which is observed with RCI and Synapse but not with AdaPlanner. LMAs still fail to select correct action types (AdaPlanner) or XPath (Synapse), and they also predict unnecessary actions (RCI).

## 6.3 DO ADVANCED LLMs SOLVE COMPOSITIONAL TASKS?

Figure 4 presents the results when we adopt other advanced LLMs, than `gpt-3.5-turbo`, as a backbone of each LMA. The more capable models, such as `gpt-4` in a generalist aspect and `text-davinci-003` in a code generation, can improve the success rate of all the prompted LMAs. However, even `gpt-4` is still far from the generalization in compositional tasks (from 28.7% to 56.0% by RCI) or from dealing with reverse-order instructions (from 19.2% to 43.5% by RCI). This indicates that we need much better foundation models to realize deployable LMA in complex real-world decision-making tasks. We provide failure examples in Appendix G.

## 6.4 WHAT DETERMINES TASK COMPLEXITY IN WEB AUTOMATION?

Figure 5 visualizes the correlation between the success rate averaged across WebGUM, HTML-T5, RCI, AdaPlanner, and Synapse (y-axis) and each statistic of compositional tasks (x-axis), such as synthesized success rate – a product of base task success rates among compositional tasks – the number of instruction tokens, and max depth of HTML subtrees. Synthesized success rate positively correlates with an average success rate ($R = 0.691$), indicating that compositional task difficulty takes over base task difficulties. In addition, the number of instruction tokens ($R = -0.579$) and the max depth of HTML subtrees ($R = -0.433$) show negative correlations. All those are statistically significant in paired t-test with $p < 0.01$. In contrast, other task statistics, such as synthesized success rate with human performance, the number of HTML tokens, and elements in HTML, just show relatively weaker correlations (see Appendix F for the details). This analysis suggests that HTML with larger depth and long instructions make generalizing compositional tasks challenging. The complexity of HTML is determined by its depth rather than its length or the number of elements. This might come from the hierarchical nature of HTML: in deeper HTML subtrees, the elements near the root tend to be distant from each other after the traversal. Such sparsity may cause confusion during planning.

| RCI (Kim et al., 2023) | | AdaPlanner (Sun et al., 2023) | | Synapse (Zheng et al., 2023) | |
|---|---|---|---|---|---|
| *Select rJ and click Submit, after clicking on the "yes" button* | | *Select OkRi7, and click Submit, after clicking on the "previous" button* | | *Select 2ld1 and click Submit, after entering the password "Zy4XI" into both text fields* | |
| ✔ | ✘ | ✔ | ✘ | ✔ | ✘ |
| 1. click //button[text()="yes"] | 1. click //*[text()="rj"]/input | 1. click //*[text()="previous"] | 1. click //*[text()="previous"] | 1. click //*[@type="password"] | 1. click //*[text()="2ld1"]/input |
| 2. click //*[text()="rj"]/input | 2. click //button[text()="yes"] | 2. click //*[text()="OkRi7"]/input | 2. type OkRi7 | 2. type Zy4XI | 2. click //*[@type="password"][1] |
| 3. click //*[@id="subbtn"] | 3. type rj | 3. click //*[@id="subbtn"] | 3. click //*[@id="subbtn"] | 3. click //*[text()="verify"] | 3. type Zy4XI |
| | 4. click //*[@id="subbtn"] | | | 4. type Zy4XI | 4. click //*[@type="password"][2] |
| | | | | 5. click //*[text()="2ld1"]/input | 5. type Zy4XI |
| | | | | 6. click //*[@id="subbtn"] | 6. click //*[@id="subbtn"] |

Table 3: Failure examples in CompWoB with **reverse-order** instructions. LMAs often fail to parse the instruction into the correct-order plan, and hallucinate unnecessary actions (e.g. *type*).

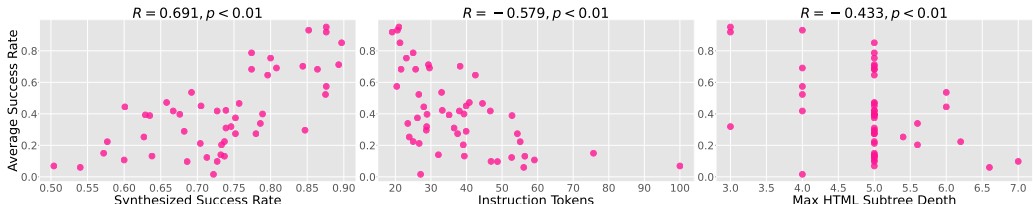

Figure 5: 2D-scatter plots between success rate averaged among LMAs (y-axis) and each statistic of compositional task (x-axis), such as success rate synthesized with a product of base task success rate, the number of instruction tokens, and max depth of HTML subtrees. Synthesized success rate positively correlates with an average success rate ($R = 0.691$, statistically significant in paired t-test with $p < 0.01$), indicating that base task difficulty may determine compositional task difficulty. In addition, the number of instruction tokens ($R = -0.579$; $p < 0.01$) and the max depth of HTML subtrees ($R = -0.433$; $p < 0.01$) show negative correlations, which suggests the high complexity of observation and long instructions make the compositional tasks hard to resolve.

## 7 DISCUSSION

**Generalizable Prompting Methods** The results of Synapse and RCI in Figure 2 imply that those prompted LMAs have some "over-fitting" trends to the base MiniWoB tasks. While the robustness across the prompts has been investigated in natural language tasks (Wei et al., 2022; Kojima et al., 2022), it is not well understood in the decision making problems. Because we will not be able to prepare the optimal self-improvement iterations or decomposed prompts for all the possible instructions and task compositions, even if using optimal exemplar retrievers, we should care more about the generalization of prompting methods for the agent systems.

**Agent-Specialized Large Language Models** As shown in Figure 4, the more capable LLMs, such as gpt-4, can improve the performance of LMAs in CompWoB. However, it has not reached the base MiniWoB yet (e.g. from 90.6% to 56.0% in RCI, and from 98.5% to 41.4% in Synapse). Similarly, as described in Section 4.4, transferred LMAs can perform better if the training dataset has a good balance and coverage, but it is far from sufficient compositional generalization or instruction generalization. The current pre-trained LLMs may still not be sufficient to generalize to complex decision making tasks, and then, in addition to prompting methods, the development of agent-specialized LLMs with enhanced reasoning and generalization would be expected.

**Parsing Complex Instructions to Executable Plan** Section 6.2 highlights that LMAs are fragile when we increase the complexity of instruction even by the most straightforward reverse-order instructions. This may not be preferable for the real-world application since the instructions might not be easy-to-parse and the users should carefully and concisely tell what they would like to do, which hinders the user's experience. It would be an interesting future direction to investigate better planning modules that could parse complex instructions to correct and executable plans.

## 8 CONCLUSION

The robustness and generalization of LMAs are important aspects for real-world deployment. We extensively examine how much existing LMAs, via transferring and prompting, can deal with a set of compositional web automation environments, CompWoB, that consists of easily-resolvable base primitive tasks. Our evaluation implies the contrary conclusion to the prior works (Table 4); the prompted LMAs are strong solver for primitive web automation tasks but significantly drop their performance in unknown task compositionality. The transferred LMAs often show sub-optimal performance in basic tasks but can deal with compositional problems much better. Our detailed analysis also highlights that LMAs also face catastrophic degradation when they receive complex, even in the simplest reversed-order instructions, and that the challenges in compositional tasks might come from instruction length and the depth of HTML subtree. We hope this inspires the community to build robust and generalizable LMAs to task compositionality toward real-world application.

| | Base MiniWoB | CompWoB | Reverse-Order Instructions | Advanced Models |
|---|---|---|---|---|
| Prompted LMA | **94.0**% / **98.5**% | 24.9% / 28.7% | 18.0% / 19.2% | 42.3% / 56.0% |
| Transferred LMA | 85.4% / 95.2% | **54.8**% / **61.5**% | **31.0**% / **34.3**% | – |

Table 4: Summary of average / max success rate in web automation.

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

APPENDIX

## A  DETAILS OF LLM API

We used OpenAI API to call LLM inference in our experiments. Table 5 shows the API used for each method. We did most of our experiments from 2023/07 to 2023/09. We use the official implementations and prompts released by the authors [2][3][4]. We spent about \$3.6K for the experiments in total.

| Methods | API | Cost (input/output; /1K tokens) | Context Length |
|---|---|---|---|
| RCI (Kim et al., 2023) | gpt-3.5-turbo | \$0.0015 / \$0.002 | 4K tokens |
| | gpt-4 | \$0.03 / \$0.06 | 8K tokens |
| AdaPlanner (Sun et al., 2023) | gpt-3.5-turbo | \$0.0015 / \$0.002 | 4K tokens |
| | text-davinci-003 | \$0.02 / \$0.02 | 4K tokens |
| Synapse (Zheng et al., 2023) | gpt-3.5-turbo | \$0.0015 / \$0.002 | 4K tokens |
| | gpt-4 | \$0.03 / \$0.06 | 8K tokens |

Table 5: List of LLM API used in this paper. We did those experiments from 2023/07 to 2023/09.

## B  PSEUDO CODE FOR PROMPTED LANGUAGE MODEL AGENTS

---

**Algorithm 1** Prompted Language Model Agents: RCI, AdaPlanner, Synapse

---

**Input:** prompt $P$, LMA $\pi$, task $\psi$, environment Env, large language model LLM, number of ERCI $N_{\text{ERCI}}$, number of IRCI $N_{\text{IRCI}}$

1: $s, g \leftarrow$ Env.reset$(\psi)$
2: $s, g \leftarrow$ LLM$(\cdot|P_{\text{syn}}, s, g)$           ▷ Task Reformulation (Synapse)
3: history $\leftarrow \{\}$
4: **while** Env is not terminated **do**
5:     $\{a_1, ..., a_T\} \leftarrow \pi(\cdot|P_\pi, s, g)$           ▷ Planning
6:     **for** $i$ **in** range$(N_{\text{ERCI}})$ **do**
7:        criticism $\leftarrow$ LLM$(\cdot|P_{\text{rci}}, \{a_1, ..., a_T\})$           ▷ Criticism (RCI)
8:        $\{a_1, ..., a_T\} \leftarrow \pi(\cdot|P_\pi, s, g, \{a_1, ..., a_T\}, \text{criticism})$           ▷ Improvement (RCI)
9:     **end for**
10:     **for** $a$ **in** $\{a_1, ..., a_T\}$ **do**
11:        **for** $j$ **in** range$(N_{\text{IRCI}})$ **do**
12:           $a \leftarrow \pi(\cdot|P_\pi, s, g, \{a, ..., a_T\}, \text{history})$           ▷ Improvement (RCI)
13:        **end for**
14:        $s, r, \text{info} \leftarrow$ Env.step(a)
15:        $\{a, ..., a_T\} \leftarrow \pi(\cdot|P_\pi, s, g, \{a, ..., a_T\}, \text{history}, \text{info})$           ▷ Replanning (AdaPlanner)
16:        history $\leftarrow$ history $\cup \{a\}$
17:     **end for**
18: **end while**

---

[2] https://github.com/posgnu/rci-agent
[3] https://github.com/haotiansun14/AdaPlanner
[4] https://github.com/ltzheng/Synapse

## C  DETAILS OF HYPER-PARAMETERS

### C.1  RCI

As we described in Section 4.1, RCI has two important hyper-parameters to control the number of self-improvement iterations. In Explicit RCI (ERCI) loop, LLMs criticize their own generated plans to identify the problem and then improve it, reflecting self-criticism. In Implicit RCI (IRCI) loop, LLMs ground the action to the current state (i.e. HTML) and refine its formatting to be parsable without self-criticism, which may reduce hallucinations or tiny errors. We here test how many self-improvement loops RCI requires (IRCI: 1-4, ERCI: 0-2). Table 6 shows that the optimal number of inference loops is different among tasks, while the recommendations are ERCI = 1 and IRCI = 3. These two hyper-parameters might need to be adjusted for each task.

| Tasks | (ERCI, IRCI) | | | | | | | | | | | |
|---|---|---|---|---|---|---|---|---|---|---|---|---|
| | (0,1) | (0,2) | (0,3) | (0,4) | (1,1) | (1,2) | **(1,3)** | (1,4) | (2,1) | (2,2) | (2,3) | (2,4) |
| click-button | **1.00** | **1.00** | **1.00** | **1.00** | 0.92 | 0.84 | 0.87 | 0.88 | 0.93 | 0.86 | 0.87 | 0.87 |
| click-checkboxes | 0.90 | 0.94 | 0.87 | 0.91 | 0.96 | 0.94 | 0.97 | **1.00** | 0.89 | 0.91 | 0.94 | 0.99 |
| click-dialog | **1.00** | **1.00** | **1.00** | **1.00** | **1.00** | **1.00** | **1.00** | **1.00** | **1.00** | **1.00** | **1.00** | **1.00** |
| click-link | 0.96 | 0.95 | 0.98 | **0.99** | 0.91 | 0.91 | 0.89 | 0.88 | 0.95 | 0.91 | 0.91 | 0.87 |
| click-option | 0.82 | 0.77 | 0.79 | **0.87** | 0.41 | 0.54 | 0.56 | 0.52 | 0.83 | 0.82 | 0.73 | **0.87** |
| click-scroll-list | **0.86** | 0.86 | 0.84 | 0.83 | 0.75 | 0.81 | 0.78 | 0.79 | 0.76 | 0.81 | 0.85 | 0.74 |

Table 6: The success rate of RCI with different hyper-parameters. The optimal parameters differ in each task, while the recommended one is (1,3).

### C.2  ADAPLANNER

We use the demonstrations of these 13 tasks where they are included in the task composition:

- enter-text
- click-widget
- navigate-tree
- login-user-popup
- email-inbox-forward-nl-turk
- click-checkboxes-large
- click-tab-2-hard
- click-dialog-2
- search-engine
- click-checkboxes-soft
- use-autocomplete
- enter-date
- click-dialog-2

### C.3  SYNAPSE

As we explained in Section 4.3, Synapse has several hyper-parameters to construct optimal structured prompts per task to specify whether LLMs translate the instruction or HTML.

Table 7 summarizes the type of reformulation into 7 categories and clarifies which transformed inputs are used for predicting open-loop plans. For instance, `Task` only requires translated instructions (and few-shot planning exemplars), although `Obs` takes raw instruction, HTML, and translated HTML as inputs. For the tasks that require temporal abstraction, it also employs state-conditional decomposition, which factorizes demonstrations into a set of exemplars conditioned on the environmental states, and can reduce error accumulation over the time step.

Table 8 provides the detailed values for state-filtering and task reformulation, which is quite different across the tasks. These well-designed structured prompts could be the source of the best performance in base MiniWoB. However, in compositional settings, it is challenging to modify them for any combinations. Instead, we assume the optimal retriever always picks up the exemplars for one of the base tasks, and we compute the maximum score among the results with given prompts.

| | Reformulation Strategies | | | | | | |
|---|---|---|---|---|---|---|---|
| **Inputs** | Task | Obs | Obs_Task | Obs_Task_Filter | Raw_Task | None | None_Filter |
| **Instruction** | Translated | Raw | Translated | Translated | Raw | Raw | Raw |
| **HTML** | ✗ | Raw+Translated | Raw+Translated | Translated | ✗ | Raw | Translated |

Table 7: Summary of task reformulation for structured prompting used in Synapse (Zheng et al., 2023). Structured prompts are finely designed per task.

| Tasks | State Filtering | Exemplar Decomposition | Raw Task Only | Reformulation |
|---|---|---|---|---|
| book-flight | True | True | False | None |
| choose-date | False | False | False | Task |
| choose-list | False | False | True | None |
| click-button | False | False | False | None |
| click-button-sequence | False | False | False | None |
| click-checkboxes | False | False | True | None |
| click-checkboxes-large | False | False | True | None |
| click-checkboxes-soft | False | False | False | Obs |
| click-checkboxes-transfer | False | False | True | None |
| click-collapsible | False | False | True | None |
| click-collapsible-2 | False | False | False | Obs Task |
| click-color | False | False | True | None |
| click-dialog | False | False | True | None |
| click-dialog-2 | False | False | False | None |
| click-link | False | False | True | None |
| click-menu | False | False | True | Task |
| click-option | False | False | True | None |
| click-pie | False | False | True | None |
| click-scroll-list | False | False | True | None |
| click-shades | False | False | True | Task |
| click-shape | True | False | False | Obs Task |
| click-tab | False | False | True | None |
| click-tab-2 | True | False | False | Obs Task |
| click-tab-2-hard | True | False | False | Obs Task |
| click-test | False | False | False | None |
| click-test-2 | False | False | False | None |
| click-widget | False | False | True | None |
| copy-paste | False | False | False | None |
| copy-paste-2 | False | False | False | None |
| count-shape | True | False | False | Obs Task |
| email-inbox | False | False | False | Task |
| email-inbox-forward-nl | False | False | False | Task |
| email-inbox-forward-nl-turk | False | False | False | Task |
| email-inbox-nl-turk | False | False | False | Task |
| enter-date | False | False | True | None |
| enter-password | False | False | True | None |
| enter-text | False | False | True | None |
| enter-text-dynamic | False | False | True | None |
| enter-time | False | False | True | None |
| find-word | True | False | False | None |
| focus-text | False | False | True | None |
| focus-text-2 | False | False | True | None |
| grid-coordinate | False | False | True | None |
| guess-number | False | False | False | None |
| identify-shape | False | False | False | None |
| login-user | False | False | True | None |
| login-user-popup | False | False | True | None |
| multi-layouts | False | False | False | None |
| multi-orderings | False | False | False | None |
| navigate-tree | False | False | False | None |
| read-table | False | False | False | None |
| search-engine | False | False | True | None |
| simple-algebra | False | False | False | None |
| simple-arithmetic | False | False | False | None |
| social-media | False | False | False | Obs Task |
| social-media-all | False | False | True | None |
| social-media-some | False | False | True | None |
| terminal | False | True | False | None |
| text-transform | False | False | False | None |
| tic-tac-toe | True | False | False | Obs Task |
| unicode-test | False | False | False | None |
| use-autocomplete | False | True | False | None |
| use-spinner | False | False | False | Task |

Table 8: Hyperparameters for Synapse (Zheng et al., 2023). **Raw Task Only** is specified with **Task as Reformation** flag, and **Reformulation** is specified with **Reformat Input** flag in the original imprementation.

# D  RANKING BASE MINIWOB TASKS

To ensure the solvability of CompWoB to some extent and to identify the data-redundant tasks for finetuned LMAs, we estimate the brute-force task difficulty (Furuta et al., 2021) (Table 9). We compute the average success rate for each task across representative previous web automation agents, such as CC-Net (SL, SL+RL) (Humphreys et al., 2022), WGE (Liu et al., 2018), WebN-T5 (Gur et al., 2022), WebGUM (Furuta et al., 2023), HTML-T5 (Gur et al., 2023), RCI (Kim et al., 2023), AdaPlanner (Sun et al., 2023), Pix2Act (BC, RL) (Shaw et al., 2023), and Synapse (Zheng et al., 2023). Based on those proxy difficulty measures, we classify 65 tasks into three categories (Kim et al., 2023): easy (from 0.8 to 1.0), medium (from 0.6 to 0.8), and hard (from 0.0 to 0.6).

| Category | Task | Task Difficulty |
|---|---|---|
| easy (0.8 - 1.0) | click-button | 0.923 |
| | click-button-sequence | 0.954 |
| | click-checkboxes | 0.936 |
| | click-checkboxes-transfer | 0.862 |
| | click-collapsible | 0.878 |
| | click-dialog | 0.923 |
| | click-link | 0.949 |
| | click-option | 0.839 |
| | click-tab | 0.898 |
| | click-test | 1.000 |
| | click-test-2 | 0.996 |
| | click-widget | 0.945 |
| | email-inbox-forward-nl | 0.844 |
| | email-inbox-forward-nl-turk | 0.804 |
| | enter-password | 0.896 |
| | enter-text | 0.922 |
| | enter-text-dynamic | 0.933 |
| | focus-text | 0.999 |
| | focus-text-2 | 0.990 |
| | grid-coordinate | 0.920 |
| | identify-shape | 0.918 |
| | login-user | 0.881 |
| | login-user-popup | 0.818 |
| | multi-layouts | 0.832 |
| | navigate-tree | 0.825 |
| | unicode-test | 0.900 |
| medium (0.6 - 0.8) | choose-date-easy | 0.740 |
| | click-checkboxes-large | 0.784 |
| | click-checkboxes-soft | 0.754 |
| | click-collapsible-2 | 0.693 |
| | click-color | 0.742 |
| | click-dialog-2 | 0.780 |
| | click-menu | 0.607 |
| | click-pie | 0.769 |
| | click-shape | 0.664 |
| | click-tab-2 | 0.736 |
| | click-tab-2-hard | 0.651 |
| | copy-paste | 0.610 |
| | email-inbox | 0.778 |
| | email-inbox-nl-turk | 0.779 |
| | enter-date | 0.714 |
| | multi-orderings | 0.793 |
| | read-table | 0.660 |
| | search-engine | 0.723 |
| | simple-algebra | 0.799 |
| | simple-arithmetic | 0.782 |
| | social-media | 0.733 |
| | text-transform | 0.737 |
| | use-autocomplete | 0.782 |
| hard (0.0 - 0.6) | book-flight | 0.510 |
| | choose-date | 0.331 |
| | choose-date-medium | 0.497 |
| | choose-list | 0.520 |
| | click-scroll-list | 0.401 |
| | click-shades | 0.501 |
| | copy-paste-2 | 0.547 |
| | count-shape | 0.536 |
| | enter-time | 0.521 |
| | find-word | 0.590 |
| | guess-number | 0.363 |
| | social-media-all | 0.432 |
| | social-media-some | 0.532 |
| | terminal | 0.592 |
| | tic-tac-toe | 0.598 |
| | use-spinner | 0.457 |

Table 9: Brute-force task complexity and difficulty classification of MiniWoB. We split 65 tasks into the three category based on the task complexity: easy (0.8 - 1.0), medium (0.6 - 0.8), and hard (0.0 - 0.6).

# E    DETAILS OF MINIWOB DATASET

To resolve the data-imbalance problem, we first run Synapse (Zheng et al., 2023) on MiniWoB and collect 77K additional demonstrations across 16 tasks on top of 347K demonstrations (Furuta et al., 2023) to compensate for the lack of data in specific tasks (**Strategy A**). We use PaLM 2-L (Anil et al., 2023) as a backbone LLM for Synapse. We then reduce the number of demonstrations for the tasks the agents can solve to focus on more challenging tasks. Based on brute-force task complexity (Appendix D), we consider the following four strategies:

- Removing 50% episodes from top-10 `easy` tasks (**Strategy B**; -63K)
- Removing 80% episodes from top-10 `easy` tasks (**Strategy C**; -92K)
- Removing 50% episodes from `easy` tasks (**Strategy D**; -132K)
- Removing 80% episodes from top-15 `easy` tasks and removing 50% episodes from other 11 `easy` tasks (**Strategy E**; -165K)

Through the empirical evaluations (Section 4.4), we find that **Strategy D** realizes a well-balanced dataset to improve the performance.

| Task | Original (347K) | Strat. A (424K) | Strat. B (361K) | Strat. C (332K) | Strat. D (292K) | Strat. E (259K) |
|---|---|---|---|---|---|---|
| book-flight | 2.88% | 2.49% | 2.84% | 3.11% | 3.54% | 4.15% |
| choose-date | 0.11% | 1.25% | 1.42% | 1.55% | 1.77% | 2.07% |
| choose-date-easy | 0.97% | 0.84% | 0.95% | 1.04% | 1.19% | 1.39% |
| choose-date-medium | 0.64% | 0.55% | 0.63% | 0.69% | 0.79% | 0.92% |
| choose-list | 0.54% | 1.24% | 1.42% | 1.55% | 1.77% | 2.07% |
| click-button | 2.82% | 2.44% | 1.39% | 0.61% | 1.73% | 0.81% |
| click-button-sequence | 2.88% | 2.49% | 1.42% | 0.62% | 1.77% | 0.83% |
| click-checkboxes | 2.81% | 2.43% | 1.36% | 0.55% | 1.69% | 0.73% |
| click-checkboxes-large | 0.57% | 2.15% | 2.46% | 2.49% | 3.06% | 3.58% |
| click-checkboxes-soft | 2.66% | 2.30% | 2.63% | 2.87% | 3.27% | 3.83% |
| click-checkboxes-transfer | 2.88% | 2.49% | 2.85% | 3.11% | 1.77% | 2.07% |
| click-collapsible | 1.71% | 1.48% | 1.69% | 1.85% | 1.06% | 1.24% |
| click-collapsible-2 | 0.63% | 0.55% | 0.63% | 0.68% | 0.78% | 0.91% |
| click-color | 0.74% | 1.23% | 1.40% | 1.53% | 1.74% | 2.04% |
| click-dialog | 2.88% | 2.49% | 2.85% | 3.11% | 1.77% | 0.83% |
| click-dialog-2 | 0.95% | 1.36% | 1.55% | 1.70% | 1.93% | 2.26% |
| click-link | 2.87% | 2.48% | 1.42% | 0.62% | 1.76% | 0.83% |
| click-menu | 0.93% | 1.24% | 1.42% | 1.55% | 1.77% | 2.07% |
| click-option | 2.88% | 2.49% | 2.84% | 3.11% | 1.77% | 2.07% |
| click-pie | 1.07% | 2.32% | 2.65% | 2.90% | 3.30% | 3.86% |
| click-scroll-list | 0.00% | 1.01% | 1.16% | 1.26% | 1.44% | 1.69% |
| click-shades | 0.00% | 1.25% | 1.42% | 1.55% | 1.77% | 2.07% |
| click-shape | 1.76% | 1.80% | 2.05% | 2.24% | 2.56% | 2.99% |
| click-tab | 2.88% | 2.49% | 2.84% | 3.10% | 1.77% | 2.07% |
| click-tab-2 | 0.53% | 0.46% | 0.52% | 0.57% | 0.65% | 0.76% |
| click-tab-2-hard | 0.45% | 1.67% | 1.91% | 2.09% | 2.38% | 2.78% |
| click-test | 2.88% | 2.49% | 1.42% | 0.62% | 1.77% | 0.83% |
| click-test-2 | 2.88% | 2.49% | 1.42% | 0.62% | 1.77% | 0.83% |
| click-widget | 2.87% | 2.48% | 1.41% | 0.62% | 1.76% | 0.82% |
| count-shape | 1.69% | 1.10% | 1.26% | 1.37% | 1.56% | 1.83% |
| email-inbox | 1.49% | 1.29% | 1.47% | 1.60% | 1.83% | 2.14% |
| email-inbox-forward-nl | 2.88% | 2.49% | 2.84% | 3.11% | 1.77% | 2.07% |
| email-inbox-forward-nl-turk | 1.41% | 1.22% | 1.39% | 1.52% | 0.86% | 1.01% |
| email-inbox-nl-turk | 1.25% | 1.08% | 1.24% | 1.35% | 1.54% | 1.80% |
| enter-date | 2.88% | 2.49% | 2.85% | 3.11% | 3.54% | 4.15% |
| enter-password | 2.88% | 2.49% | 2.84% | 3.10% | 1.77% | 2.07% |
| enter-text | 2.88% | 2.49% | 2.85% | 3.11% | 1.77% | 0.83% |
| enter-text-dynamic | 2.88% | 2.49% | 1.42% | 0.62% | 1.77% | 0.83% |
| enter-time | 0.00% | 1.08% | 1.23% | 1.35% | 1.54% | 1.80% |
| focus-text | 2.88% | 2.49% | 1.42% | 0.62% | 1.77% | 0.83% |
| focus-text-2 | 2.88% | 2.49% | 1.42% | 0.62% | 1.77% | 0.83% |
| grid-coordinate | 2.41% | 2.08% | 2.38% | 2.60% | 1.41% | 0.55% |
| guess-number | 0.29% | 1.25% | 1.42% | 1.55% | 1.77% | 2.07% |
| identify-shape | 2.60% | 2.24% | 2.56% | 2.80% | 1.60% | 0.75% |
| login-user | 2.82% | 2.44% | 2.79% | 3.05% | 1.73% | 2.03% |
| login-user-popup | 2.82% | 2.44% | 2.78% | 3.04% | 1.73% | 2.03% |
| multi-layouts | 2.88% | 2.49% | 2.85% | 3.11% | 1.77% | 2.07% |
| multi-orderings | 2.88% | 2.49% | 2.85% | 3.11% | 3.54% | 4.15% |
| navigate-tree | 2.84% | 2.46% | 2.81% | 3.07% | 1.73% | 2.02% |
| search-engine | 2.56% | 2.21% | 2.52% | 2.76% | 3.14% | 3.68% |
| social-media | 0.76& | 0.66% | 0.75% | 0.82% | 0.93% | 1.09% |
| social-media-all | 0.03% | 0.02% | 0.03% | 0.03% | 0.03% | 0.04% |
| social-media-some | 0.09& | 0.08% | 0.09% | 0.10% | 0.11% | 0.13% |
| tic-tac-toe | 1.14% | 1.43% | 1.63% | 1.78% | 2.03% | 2.37% |
| use-autocomplete | 1.00% | 0.86% | 0.99% | 1.08% | 1.23% | 1.44% |
| use-spinner | 0.15% | 1.19% | 1.36% | 1.48% | 1.69% | 1.98% |

Table 10: Task ratio in rebalanced dataset for HTML-T5++.

## F    TASK COMPLEXITY ANALYSIS IN WEB AUTOMATION

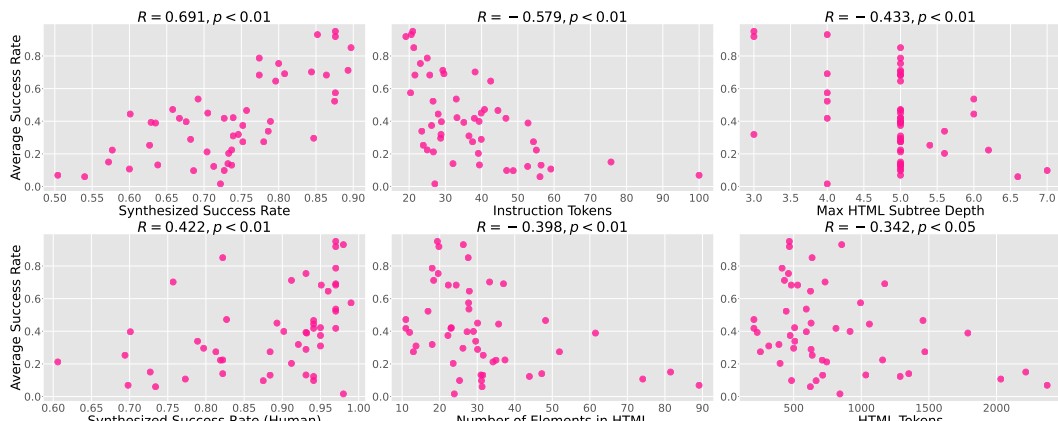

Figure 6: 2D-scatter plots between success rate averaged among LMAs (y-axis) and each statistic of compositional task (x-axis), such as success rate synthesized with a product of base task success rate, the number of instruction tokens, max depth of HTML subtrees, success rate synthesized with a product of base task human performances, the number of elements in HTML, and the number of HTML tokens. Synthesized success rate positively correlates with an average success rate ($R = 0.691$, statistically significant in paired t-test with $p < 0.01$), indicating that base task difficulty may determine compositional task difficulty. In addition, the number of instruction tokens ($R = -0.579$; $p < 0.01$) and the max depth of HTML subtrees ($R = -0.433$; $p < 0.01$) show negative correlations, which suggests the high complexity of observation and long instructions make the compositional tasks hard to resolve. In contrast, synthesized success rate from human performance, the number of HTML tokens, and elements in HTML just show relatively weaker correlations.

## G    FAILURE EXAMPLES WITH ADVANCED LANGUAGE MODELS

We provide several failure episodes with advanced language models such as `gpt-4` and `text-davinci-003` in Table 11. The left columns have correct plans, and the right columns have failure plans. Compared to `gpt-3.5-turbo` (Table 2), LMAs based on advanced models may not ignore the necessary intermediate actions. However, they tend to suffer from tiny errors such as capitalization (RCI, AdaPlanner) or attributes in HTML (Synapse).

| RCI (Kim et al., 2023) | | AdaPlanner (Sun et al., 2023) | | Synapse (Zheng et al., 2023) | |
|---|---|---|---|---|---|
| *Click on the link "adipiscing", and then click on the "submit" button* | | Click on the link "Augue", and then click on a "button" widget | | Select 0Qm9EUt, and then enter the username "cristin" and the password "M5" into the text fields and press login | |
| ✔ | ✗ | ✔ | ✗ | ✔ | ✗ |
| 1. click //span[text()="adipiscing"] 
 2. click //button[text()="submit"] | 1. click //span[text()="adipiscing"] 
 2. click //button[text()="Submit"] | 1. click //span[text()="Augue"] 
 2. click //*[@data-type="button"] | 1. click //span[text()="augue"] 
 2. click //*[@data-type="button"] | 1. click //*[text()="0Qm9EUt"]/input 
 2. click //*[@id="username"] 
 3. type cristin 
 4. click //*[@id="password"] 
 5. type M5 
 6. click //*[@id="subbtn"] | 1. click //*[@id="0Qm9EUt"] 
 2. click //*[@id="username"] 
 3. type cristin 
 4. click //*[@id="password"] 
 5. type M5 
 6. click //*[@id="subbtn"] |

Table 11: Failure examples in CompWoB with advanced models (gpt-4, text-davinci-003).

# H   PER-TASK PERFORMANCE ON MINIWOB AND COMPWOB

| Task | HTML-T5 (Gur et al., 2023) | HTML-T5++ (Ours) |
|------|---------------------------|------------------|
| book-flight | 0.99 | 0.99 |
| choose-date | 0.16 | 1.00 |
| choose-date-easy | 1.00 | 1.00 |
| choose-date-medium | 0.56 | 1.00 |
| choose-list | 0.22 | 1.00 |
| click-button | 1.00 | 1.00 |
| click-button-sequence | 1.00 | 1.00 |
| click-checkboxes | 1.00 | 1.00 |
| click-checkboxes-large | 0.90 | 0.97 |
| click-checkboxes-soft | 0.99 | 1.00 |
| click-checkboxes-transfer | 1.00 | 1.00 |
| click-collapsible | 1.00 | 1.00 |
| click-collapsible-2 | 0.93 | 0.96 |
| click-color | 1.00 | 1.00 |
| click-dialog | 1.00 | 1.00 |
| click-dialog-2 | 0.74 | 1.00 |
| click-link | 0.99 | 1.00 |
| click-menu | 0.37 | 0.96 |
| click-option | 1.00 | 1.00 |
| click-pie | 0.96 | 0.94 |
| click-scroll-list | 0.99 | 1.00 |
| click-shades | 0.00 | 1.00 |
| click-shape | 0.79 | 0.95 |
| click-tab | 1.00 | 1.00 |
| click-tab-2 | 0.94 | 0.98 |
| click-tab-2-hard | 0.88 | 0.96 |
| click-test | 1.00 | 1.00 |
| click-test-2 | 1.00 | 1.00 |
| click-widget | 1.00 | 1.00 |
| count-shape | 0.67 | 0.92 |
| email-inbox | 1.00 | 0.98 |
| email-inbox-forward-nl | 1.00 | 1.00 |
| email-inbox-forward-nl-turk | 1.00 | 1.00 |
| email-inbox-nl-turk | 0.99 | 1.00 |
| enter-date | 1.00 | 1.00 |
| enter-password | 1.00 | 1.00 |
| enter-text | 1.00 | 1.00 |
| enter-text-dynamic | 1.00 | 1.00 |
| enter-time | 1.00 | 1.00 |
| focus-text | 1.00 | 1.00 |
| focus-text-2 | 1.00 | 1.00 |
| grid-coordinate | 1.00 | 1.00 |
| guess-number | 0.13 | 1.00 |
| identify-shape | 1.00 | 1.00 |
| login-user | 1.00 | 1.00 |
| login-user-popup | 1.00 | 1.00 |
| multi-layouts | 1.00 | 1.00 |
| multi-orderings | 1.00 | 1.00 |
| navigate-tree | 0.99 | 1.00 |
| search-engine | 0.93 | 0.96 |
| social-media | 0.99 | 1.00 |
| social-media-all | 0.31 | 0.31 |
| social-media-some | 0.89 | 0.85 |
| tic-tac-toe | 0.57 | 0.55 |
| use-autocomplete | 0.97 | 0.99 |
| use-spinner | 0.07 | 0.06 |
| **Average** | **0.856** | **0.952** |

Table 12: Per-task average success rate on 56 tasks from MiniWoB++. We refer to Gur et al. (2023) for the baseline performance.

| Task | WebGUM | HTML-T5 | HTML-T5++ | RCI (zero) | RCI (first) | RCI (second) | RCI (comb) | RCI (gpt-4) | Synapse (first) | Synapse (second) | Synapse (best) | Synapse (fgpt-4) | Synapse (sgpt-4) | Synapse (bgpt-4) | AdaPlanner | AdaPlanner (davinci-3) |
|---|---|---|---|---|---|---|---|---|---|---|---|---|---|---|---|---|
| click-button_click-checkboxes | 0.210 | 0.700 | 0.790 | 0.370 | 0.710 | 0.730 | 0.800 | 0.880 | 0.810 | 0.840 | 0.840 | 0.980 | 0.840 | 0.980 | 0.060 | 0.890 |
| click-button_click-checkboxes-transfer | 0.010 | 0.600 | 0.750 | 0.270 | 0.790 | 0.790 | 0.810 | 1.000 | 0.540 | 0.740 | 0.740 | 0.960 | 0.740 | 0.960 | 0.000 | 0.940 |
| click-button_click-dialog | 0.870 | 0.720 | 0.970 | 0.740 | 0.830 | 0.820 | 0.940 | 1.000 | 1.000 | 0.650 | 1.000 | 1.000 | 0.660 | 1.000 | 0.900 | 0.980 |
| click-button_click-link | 0.810 | 0.860 | 0.860 | 0.870 | 0.830 | 0.870 | 0.920 | 0.920 | 0.990 | 0.000 | 0.990 | 1.000 | 0.000 | 1.000 | 0.940 | 0.970 |
| click-button_click-option | 0.600 | 0.840 | 0.840 | 0.580 | 0.650 | 0.370 | 0.420 | 0.420 | 0.920 | 0.870 | 0.920 | 0.960 | 0.880 | 0.960 | 0.000 | 0.860 |
| click-button-sequence_click-checkboxes | 0.490 | 0.960 | 0.900 | 0.440 | 0.520 | 0.820 | 0.670 | 0.820 | 0.710 | 0.890 | 0.890 | 0.940 | 0.840 | 0.940 | 0.060 | 0.680 |
| click-button-sequence_click-option | 0.940 | 1.000 | 1.000 | 0.540 | 0.830 | 0.620 | 0.760 | 0.800 | 0.900 | 0.900 | 0.900 | 0.900 | 0.840 | 0.900 | 0.000 | 0.490 |
| click-link_click-button | 0.950 | 0.390 | 0.310 | 0.000 | 0.000 | 0.000 | 0.000 | 0.100 | 0.900 | 0.000 | 0.000 | 0.300 | 0.000 | 0.300 | 0.420 | 0.000 |
| click-link_click-dialog | 0.900 | 0.980 | 0.970 | 0.790 | 0.830 | 0.970 | 0.980 | 0.980 | 0.010 | 1.000 | 0.000 | 1.000 | 1.000 | 1.000 | 0.950 | 0.800 |
| click-link_click-widget | 0.930 | 0.970 | 1.000 | 0.200 | 0.200 | 0.460 | 0.600 | 0.640 | 0.000 | 0.000 | 0.000 | 0.000 | 0.000 | 0.000 | 0.540 | 0.490 |
| click-link_enter-text | 0.940 | 0.900 | 0.990 | 0.550 | 0.520 | 0.820 | 0.900 | 0.780 | 0.370 | 0.370 | 0.370 | 0.820 | 0.000 | 0.820 | 0.980 | 0.890 |
| click-option_enter-text | 0.130 | 0.900 | 1.000 | 0.090 | 0.160 | 0.200 | 0.060 | 0.840 | 0.000 | 0.000 | 0.000 | 0.000 | 0.000 | 0.000 | 0.770 | 0.920 |
| click-option_login-user | 1.000 | 0.970 | 1.000 | 0.380 | 0.130 | 0.410 | 0.500 | 0.880 | 0.000 | 0.000 | 0.000 | 1.000 | 0.000 | 1.000 | 0.000 | 0.960 |
| click-option_navigate-tree | 0.980 | 1.000 | 0.960 | 0.010 | 0.470 | 0.500 | 0.720 | 0.860 | 0.000 | 0.240 | 0.240 | 0.560 | 0.960 | 0.960 | 0.620 | 0.590 |
| click-widget_enter-password | 0.020 | 0.480 | 0.370 | 0.550 | 0.000 | 0.000 | 0.000 | 0.820 | 0.000 | 0.000 | 0.000 | 0.000 | 0.000 | 0.000 | 0.000 | 0.000 |
| click-widget_multi-layouts | 0.370 | 0.740 | 0.810 | 0.000 | 0.000 | 0.000 | 0.000 | 0.740 | 0.000 | 0.000 | 0.000 | 0.000 | 0.000 | 0.000 | 0.000 | 0.000 |
| enter-password_click-option | 0.430 | 0.400 | 0.220 | 0.000 | 0.000 | 0.000 | 0.000 | 0.240 | 0.000 | 0.760 | 0.760 | 1.000 | 1.000 | 1.000 | 0.000 | 0.000 |
| login-user_navigate-tree | 1.000 | 0.690 | 0.900 | 0.000 | 0.020 | 0.000 | 0.000 | 0.780 | 0.000 | 0.000 | 0.000 | 0.000 | 0.000 | 0.000 | 0.000 | 0.000 |
| multi-layouts_login-user | 0.000 | 0.040 | 0.140 | 0.020 | 0.010 | 0.000 | 0.000 | 0.700 | 0.720 | 0.720 | 0.720 | 1.000 | 0.000 | 0.000 | 0.010 | 0.000 |
| **Average (two-way)** | **0.579** | **0.712** | **0.739** | **0.320** | **0.375** | **0.419** | **0.469** | **0.715** | **0.349** | **0.300** | **0.469** | **0.521** | **0.346** | **0.641** | **0.356** | **0.523** |
| click-button_click-option_login-user | 0.140 | 0.520 | 0.740 | 0.000 | 0.340 | 0.000 | 0.000 | 0.120 | 0.000 | 0.000 | 0.000 | 0.940 | 0.940 | 0.940 | 0.140 | 0.000 |
| click-button-sequence_click-option_login-user | 0.300 | 0.200 | 0.770 | 0.520 | 0.740 | 0.680 | 0.840 | 0.560 | 0.720 | 0.000 | 0.720 | 0.660 | 0.660 | 0.660 | 0.000 | 0.000 |
| click-checkboxes_click-widget_click-button-sequence | 0.200 | 0.580 | 0.780 | 0.400 | 0.660 | 0.660 | 0.820 | 1.000 | 0.820 | 0.000 | 0.820 | 0.820 | 0.180 | 0.820 | 0.580 | 0.720 |
| click-checkboxes-transfer_click-link_click-button-sequence_enter-password | 0.000 | 0.010 | 0.010 | 0.000 | 0.000 | 0.000 | 0.000 | 0.660 | 0.000 | 0.000 | 0.420 | 0.000 | 0.500 | 0.500 | 0.000 | 0.000 |
| click-checkboxes-transfer_enter-password_click-widget_click-dialog | 0.000 | 1.000 | 0.000 | 0.000 | 0.000 | 0.000 | 0.000 | 0.280 | 0.000 | 0.430 | 0.420 | 0.000 | 0.400 | 0.400 | 0.000 | 0.000 |
| click-dialog_click-button-sequence_enter-password | 0.890 | 1.000 | 1.000 | 0.000 | 0.540 | 0.000 | 0.000 | 0.700 | 0.000 | 0.000 | 0.000 | 0.000 | 0.000 | 0.000 | 0.000 | 0.000 |
| click-dialog_click-checkboxes-transfer_click-widget | 0.140 | 0.690 | 0.560 | 0.140 | 0.540 | 0.340 | 0.400 | 0.680 | 0.000 | 0.600 | 0.600 | 0.000 | 0.000 | 0.620 | 0.560 | 0.580 |
| click-link_click-button_click-dialog | 0.730 | 0.940 | 0.980 | 0.240 | 0.270 | 0.480 | 0.610 | 0.600 | 0.000 | 0.600 | 0.600 | 0.020 | 0.020 | 0.020 | 0.560 | 0.580 |
| click-widget_click-option_click-dialog | 0.010 | 0.120 | 0.410 | 0.120 | 0.000 | 0.020 | 0.300 | 0.400 | 0.000 | 0.000 | 0.000 | 0.000 | 0.000 | 0.000 | 0.000 | 0.000 |
| enter-password_click-checkboxes_login-user-popup | 0.110 | 0.000 | 0.060 | 0.000 | 0.000 | 0.000 | 0.000 | 0.700 | 0.720 | 0.300 | 0.300 | 0.020 | 0.000 | 0.000 | 0.000 | 0.010 |
| **Average (three-way)** | **0.252** | **0.407** | **0.531** | **0.142** | **0.255** | **0.218** | **0.297** | **0.570** | **0.154** | **0.154** | **0.256** | **0.244** | **0.198** | **0.396** | **0.128** | **0.130** |
| click-checkboxes-transfer_multi-layouts_email-inbox-forward-nl-transition | 0.240 | 0.460 | 0.650 | 0.000 | 0.000 | 0.000 | 0.000 | 0.000 | 0.000 | 0.000 | 0.000 | 0.000 | 0.000 | 0.000 | 0.440 | 0.000 |
| click-option_login-user-transition | 0.990 | 1.000 | 0.800 | 0.000 | 0.000 | 0.000 | 0.000 | 0.000 | 0.000 | 0.000 | 0.000 | 0.000 | 0.000 | 0.000 | 0.220 | 0.220 |
| click-option_multi-layouts_click-widget_login-user-popup-transition | 0.190 | 0.220 | 0.130 | 0.000 | 0.000 | 0.000 | 0.000 | 0.000 | 0.000 | 0.000 | 0.000 | 0.000 | 0.000 | 0.000 | 0.000 | 0.000 |
| login-user-popup_email-inbox-forward-nl-turk-transition | 0.960 | 1.000 | 1.000 | 0.200 | 0.360 | 0.100 | 0.460 | 0.120 | 0.000 | 0.000 | 0.000 | 0.340 | 0.340 | 0.340 | 0.700 | 0.560 |
| **Average (n-way)** | **0.092** | **0.266** | **0.410** | **0.130** | **0.158** | **0.234** | **0.222** | **0.584** | **0.000** | **0.074** | **0.074** | **0.000** | **0.244** | **0.244** | **0.014** | **0.218** |
| click-button_click-tab-2-hard | 0.240 | 0.550 | 0.510 | 0.080 | 0.120 | 0.260 | 0.340 | 0.580 | 0.280 | 0.280 | 0.280 | 0.960 | 0.320 | 0.960 | 0.260 | 0.280 |
| click-checkboxes_use-autocomplete | 0.410 | 0.220 | 0.840 | 0.200 | 0.060 | 0.640 | 0.160 | 0.820 | 0.000 | 0.720 | 0.720 | 0.420 | 0.000 | 0.420 | 0.000 | 0.000 |
| click-checkboxes-soft_enter-password | 0.990 | 0.850 | 0.910 | 0.000 | 0.000 | 0.000 | 0.000 | 0.820 | 0.000 | 0.820 | 0.820 | 0.820 | 0.820 | 0.820 | 0.000 | 0.000 |
| click-checkboxes-soft_multi-layouts | 0.930 | 0.240 | 0.170 | 0.000 | 0.000 | 0.000 | 0.000 | 0.120 | 0.000 | 0.300 | 0.000 | 0.000 | 0.000 | 0.000 | 0.000 | 0.000 |
| click-dialog_search-engine | 0.930 | 0.970 | 0.940 | 0.100 | 0.020 | 0.000 | 0.000 | 0.920 | 0.100 | 0.060 | 0.000 | 0.180 | 0.000 | 0.180 | 0.160 | 0.100 |
| click-dialog-2_click-widget | 0.270 | 0.460 | 0.690 | 0.000 | 0.000 | 0.000 | 0.040 | 0.020 | 0.100 | 0.100 | 0.100 | 0.180 | 0.180 | 0.180 | 0.160 | 0.100 |
| click-dialog-2_login-user-popup | 0.330 | 0.440 | 0.360 | 0.080 | 0.100 | 0.000 | 0.200 | 0.060 | 0.000 | 0.000 | 0.000 | 0.000 | 0.000 | 0.000 | 0.100 | 0.140 |
| click-widget_click-checkboxes-soft | 0.120 | 0.290 | 0.460 | 0.080 | 0.100 | 0.060 | 0.200 | 0.600 | 0.000 | 0.000 | 0.000 | 0.000 | 0.000 | 0.340 | 0.000 | 0.380 |
| enter-date_login-user | 0.990 | 1.000 | 0.230 | 0.000 | 0.020 | 0.000 | 0.060 | 0.500 | 0.000 | 0.000 | 0.000 | 0.000 | 0.000 | 0.000 | 0.000 | 0.000 |
| use-autocomplete_click-dialog | 0.000 | 0.000 | 0.000 | 0.000 | 0.000 | 0.080 | 0.080 | 0.060 | 0.000 | 0.000 | 0.000 | 0.000 | 0.000 | 0.000 | 0.700 | 0.700 |
| **Average (transition)** | **0.676** | **0.728** | **0.698** | **0.000** | **0.000** | **0.040** | **0.000** | **0.116** | **0.000** | **0.000** | **0.000** | **0.068** | **0.000** | **0.068** | **0.184** | **0.200** |
| **Average (two-way, easy-medium)** | **0.521** | **0.502** | **0.511** | **0.046** | **0.040** | **0.096** | **0.088** | **0.450** | **0.038** | **0.034** | **0.038** | **0.156** | **0.114** | **0.238** | **0.090** | **0.090** |
| **Average (total)** | **0.463** | **0.566** | **0.615** | **0.179** | **0.225** | **0.258** | **0.287** | **0.560** | **0.178** | **0.154** | **0.254** | **0.295** | **0.225** | **0.414** | **0.206** | **0.295** |

Table 13: Per-task success rate on CompWoB.

| Task | WebGUM | HTML-T5 | HTML-T5++ | RCI (zero) | RCI (first) | RCI (second) | RCI (comb) | RCI (gpt-4) | Synapse (first) | Synapse (second) | Synapse (best) | AdaPlanner |
|---|---|---|---|---|---|---|---|---|---|---|---|---|
| click-button_click-checkboxes | 0.180 | 0.340 | 0.630 | 0.420 | 0.310 | 0.420 | 0.380 | 0.520 | 0.670 | 0.800 | 0.800 | 0.340 |
| click-button_click-checkboxes-transfer | 0.060 | 0.260 | 0.630 | 0.150 | 0.140 | 0.230 | 0.200 | 0.320 | 0.360 | 0.930 | 0.930 | 0.090 |
| click-button_click-dialog | 0.300 | 0.020 | 0.030 | 0.850 | 0.790 | 0.790 | 0.810 | 0.840 | 0.170 | 0.000 | 0.200 | 0.000 |
| click-button_click-link | 0.010 | 0.050 | 0.040 | 0.430 | 0.500 | 0.430 | 0.460 | 0.560 | 0.870 | 0.000 | 0.870 | 0.970 |
| click-button_click-option | 0.230 | 0.420 | 0.550 | 0.350 | 0.270 | 0.400 | 0.410 | 0.820 | 0.480 | 0.870 | 0.860 | 0.440 |
| click-button-sequence_click-checkboxes | 0.000 | 0.140 | 0.130 | 0.270 | 0.260 | 0.390 | 0.280 | 0.820 | 0.350 | 0.870 | 0.870 | 0.700 |
| click-button-sequence_click-option | 0.000 | 0.020 | 0.110 | 0.360 | 0.420 | 0.320 | 0.490 | 0.920 | 0.000 | 0.000 | 0.350 | 0.550 |
| click-button-sequence_login-user-popup | 0.000 | 0.720 | 0.020 | 0.000 | 0.000 | 0.000 | 0.000 | 0.160 | 0.000 | 0.000 | 0.000 | 0.280 |
| click-link_click-button | 0.910 | 0.750 | 0.580 | 0.840 | 0.760 | 0.800 | 0.830 | 0.920 | 0.670 | 0.670 | 0.670 | 0.950 |
| click-link_click-dialog | 0.450 | 0.050 | 0.230 | 0.330 | 0.430 | 0.470 | 0.880 | 0.660 | 0.000 | 0.000 | 0.000 | 0.200 |
| click-link_click-widget | 0.140 | 0.750 | 0.740 | 0.550 | 0.580 | 0.480 | 0.530 | 0.780 | 0.640 | 0.000 | 0.640 | 0.920 |
| click-link_enter-text | 0.000 | 0.950 | 0.910 | 0.240 | 0.280 | 0.300 | 0.310 | 0.480 | 0.000 | 0.000 | 0.000 | 0.970 |
| click-option_enter-text | 1.000 | 0.660 | 0.750 | 0.570 | 0.650 | 0.650 | 0.640 | 0.800 | 0.000 | 0.000 | 0.000 | 0.590 |
| click-option_login-user | 0.980 | 0.220 | 0.150 | 0.010 | 0.000 | 0.010 | 0.000 | 0.620 | 0.050 | 0.590 | 0.590 | 0.020 |
| click-widget_enter-password | 0.000 | 0.060 | 0.050 | 0.450 | 0.390 | 0.500 | 0.490 | 0.440 | 0.000 | 0.000 | 0.000 | 0.880 |
| click-widget_multi-layouts | 0.280 | 0.470 | 0.530 | 0.000 | 0.000 | 0.000 | 0.000 | 0.300 | 0.000 | 0.000 | 0.000 | 0.000 |
| enter-password_click-option | 0.260 | 0.310 | 0.220 | 0.000 | 0.000 | 0.000 | 0.000 | 0.440 | 0.000 | 0.700 | 0.700 | 0.020 |
| login-user_navigate-tree | 0.960 | 0.160 | 0.900 | 0.000 | 0.000 | 0.000 | 0.000 | 0.920 | 0.000 | 0.000 | 0.000 | 0.000 |
| multi-layouts_login-user | 0.000 | 0.030 | 0.080 | 0.010 | 0.000 | 0.000 | 0.000 | 0.280 | 0.210 | 0.000 | 0.210 | 0.000 |
| **Average (two-way)** | **0.288** | **0.319** | **0.364** | **0.292** | **0.289** | **0.310** | **0.336** | **0.573** | **0.230** | **0.281** | **0.385** | **0.396** |
| click-button_click-option_login-user | 0.250 | 0.140 | 0.220 | 0.020 | 0.000 | 0.000 | 0.000 | 0.200 | 0.000 | 0.000 | 0.000 | 0.220 |
| click-button-sequence_click-option_login-user | 0.000 | 0.000 | 0.220 | 0.420 | 0.800 | 0.440 | 0.740 | 0.000 | 0.000 | 0.000 | 0.000 | 0.320 |
| click-checkboxes_click-widget_click-button-sequence | 0.000 | 0.270 | 0.100 | 0.060 | 0.140 | 0.140 | 0.080 | 0.820 | 0.000 | 0.000 | 0.000 | 0.000 |
| click-checkboxes-transfer_click-button-sequence_enter-password | 0.000 | 0.000 | 0.000 | 0.000 | 0.000 | 0.000 | 0.000 | 0.340 | 0.000 | 0.000 | 0.000 | 0.000 |
| click-checkboxes-transfer_enter-password_click-dialog | 0.000 | 0.000 | 0.000 | 0.000 | 0.000 | 0.000 | 0.000 | 0.000 | 0.000 | 0.000 | 0.000 | 0.000 |
| click-dialog_click-button-sequence_enter-password | 1.000 | 0.690 | 0.040 | 0.140 | 0.340 | 0.000 | 0.360 | 0.680 | 0.000 | 0.000 | 0.000 | 0.000 |
| click-dialog_click-checkboxes-transfer_click-widget | 0.080 | 0.520 | 0.460 | 0.140 | 0.320 | 0.320 | 0.310 | 0.580 | 0.000 | 0.000 | 0.000 | 0.000 |
| click-link_click-button_click-dialog | 0.040 | 0.050 | 0.130 | 0.180 | 0.220 | 0.160 | 0.140 | 0.360 | 0.000 | 0.000 | 0.000 | 0.000 |
| click-widget_click-option_click-dialog | 0.000 | 0.000 | 0.000 | 0.120 | 0.000 | 0.000 | 0.000 | 0.120 | 0.000 | 0.000 | 0.000 | 0.000 |
| enter-password_click-checkboxes_login-user-popup | 0.000 | 0.000 | 0.000 | 0.000 | 0.000 | 0.000 | 0.000 | 0.660 | 0.000 | 0.000 | 0.000 | 0.000 |
| **Average (three-way)** | **0.137** | **0.167** | **0.117** | **0.094** | **0.182** | **0.106** | **0.163** | **0.376** | **0.000** | **0.000** | **0.000** | **0.054** |
| click-button-sequence_click-widget_click-link_click-button_click-link_click-button_click-checkboxes_click-option_click-dialog | 0.000 | 0.070 | 0.090 | 0.140 | 0.100 | 0.340 | 0.080 | 0.600 | 0.000 | 0.000 | 0.000 | 0.000 |
| click-button-sequence_click-option_click-widget_click-link_click-button_click-checkboxes_click-option_click-dialog_login-user | 0.000 | 0.000 | 0.000 | 0.000 | 0.000 | 0.000 | 0.000 | 0.180 | 0.000 | 0.000 | 0.000 | 0.000 |
| click-checkboxes_click-widget_click-button-sequence_enter-password | 0.000 | 0.090 | 0.240 | 0.220 | 0.220 | 0.120 | 0.200 | 0.320 | 0.000 | 0.000 | 0.000 | 0.000 |
| click-checkboxes-transfer_enter-password_click-dialog | 0.000 | 0.060 | 0.260 | 0.170 | 0.120 | 0.140 | 0.170 | 0.380 | 0.000 | 0.000 | 0.000 | 0.000 |
| click-link_click-button_click-checkboxes_click-button_click-checkboxes_click-option_click-dialog | 0.000 | 0.000 | 0.030 | 0.060 | 0.060 | 0.040 | 0.180 | 0.080 | 0.000 | 0.000 | 0.000 | 0.000 |
| **Average (n-way)** | **0.000** | **0.044** | **0.124** | **0.118** | **0.100** | **0.128** | **0.126** | **0.312** | **0.000** | **0.000** | **0.000** | **0.000** |
| click-checkboxes-transfer_multi-layouts_email-inbox-forward-nl-transition | 0.170 | 0.360 | 0.780 | 0.000 | 0.000 | 0.000 | 0.000 | 0.000 | 0.000 | 0.000 | 0.000 | 0.140 |
| click-option_login-user-transition | 0.990 | 0.800 | 0.650 | 0.000 | 0.000 | 0.000 | 0.000 | 0.080 | 0.000 | 0.000 | 0.000 | 0.000 |
| click-option_multi-layouts_click-widget_login-user-popup-transition | 0.070 | 0.100 | 0.030 | 0.000 | 0.000 | 0.000 | 0.000 | 0.000 | 0.000 | 0.000 | 0.000 | 0.000 |
| login-user_navigate-tree-transition | 1.000 | 1.000 | 0.990 | 0.000 | 0.000 | 0.000 | 0.000 | 0.080 | 0.000 | 0.000 | 0.000 | 0.000 |
| login-user-popup_email-inbox-forward-nl-turk-transition | 0.650 | 0.770 | 0.820 | 0.000 | 0.000 | 0.220 | 0.100 | 0.300 | 0.000 | 0.000 | 0.000 | 0.580 |
| **Average (transition)** | **0.576** | **0.606** | **0.654** | **0.000** | **0.000** | **0.044** | **0.020** | **0.092** | **0.000** | **0.000** | **0.000** | **0.144** |
| click-button_click-tab-2-hard | 0.050 | 0.470 | 0.490 | 0.140 | 0.120 | 0.180 | 0.400 | 0.640 | 0.300 | 0.020 | 0.300 | 0.160 |
| click-button-sequence_use-autocomplete | 0.000 | 0.080 | 0.930 | 0.260 | 0.080 | 0.640 | 0.040 | 0.860 | 0.000 | 0.000 | 0.000 | 0.000 |
| click-checkboxes-soft_enter-password | 0.950 | 0.490 | 0.760 | 0.000 | 0.000 | 0.000 | 0.000 | 0.740 | 0.000 | 0.000 | 0.000 | 0.000 |
| click-checkboxes-soft_multi-layouts | 0.940 | 0.390 | 0.260 | 0.000 | 0.000 | 0.000 | 0.000 | 0.200 | 0.000 | 0.420 | 0.000 | 0.000 |
| click-dialog_search-engine | 0.950 | 0.820 | 0.550 | 0.000 | 0.000 | 0.020 | 0.020 | 0.980 | 0.000 | 0.000 | 0.000 | 0.080 |
| click-dialog-2_click-widget | 0.210 | 0.370 | 0.260 | 0.000 | 0.000 | 0.000 | 0.000 | 0.020 | 0.000 | 0.000 | 0.000 | 0.000 |
| click-dialog-2_login-user-popup | 0.380 | 0.360 | 0.330 | 0.000 | 0.000 | 0.000 | 0.000 | 0.020 | 0.000 | 0.000 | 0.000 | 0.000 |
| click-widget_click-checkboxes-soft | 0.090 | 0.240 | 0.290 | 0.080 | 0.080 | 0.120 | 0.080 | 0.520 | 0.000 | 0.000 | 0.000 | 0.000 |
| enter-date_login-user | 0.950 | 0.350 | 0.960 | 0.000 | 0.040 | 0.000 | 0.000 | 0.400 | 0.000 | 0.000 | 0.000 | 0.040 |
| use-autocomplete_click-dialog | 0.000 | 0.000 | 0.000 | 0.000 | 0.080 | 0.000 | 0.000 | 0.120 | 0.000 | 0.000 | 0.000 | 0.000 |
| **Average (two-way, easy-medium)** | **0.452** | **0.357** | **0.483** | **0.048** | **0.040** | **0.096** | **0.054** | **0.450** | **0.030** | **0.044** | **0.030** | **0.028** |
| **Average (total)** | **0.291** | **0.297** | **0.343** | **0.157** | **0.170** | **0.181** | **0.192** | **0.435** | **0.098** | **0.121** | **0.160** | **0.189** |

Table 14: Per-task success rate on CompWoB with reverse-order instructions.

# I   PROMPTED LANGUAGE MODEL AGENTS WITH ORACLE EXEMPLARS

We here evaluate the performance of RCI with `gpt-4` and oracle exemplars on 20 two-way tasks in CompWoB. We provide two demonstrations per task in the prompts. Figure 7 shows that RCI with `gpt-4` and oracle exemplars achieves 82.9% success rate, which is the best among the baselines, such as HTML-T5++ (73.9%), RCI with combination exemplars (`gpt-3.5-turbo`: 46.9%, `gpt-4`: 71.5%). See Table 13 for other baselines. This ensures that if a prompt includes how to perform on compositional tasks, the performance gets better. However, please also keep in mind that providing exemplars for every compositional problem is infeasible given the huge space of problems and the cost of collecting examples for every one of them.

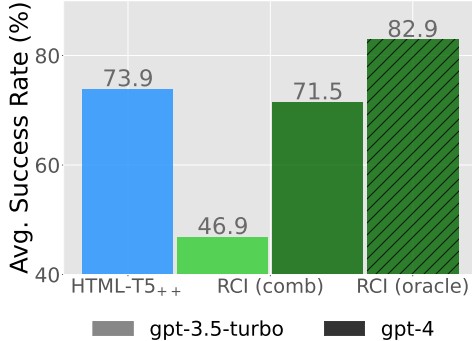

Figure 7: Average success rate of LMAs in 20 two-way tasks from CompWoB. RCI with `gpt-4` achieves the best performance when the oracle exemplars are provided (82.9%) in the prompt.

# J   TASK COMPLEXITY ANALYSIS WITH LANGUAGE MODEL AGENTS

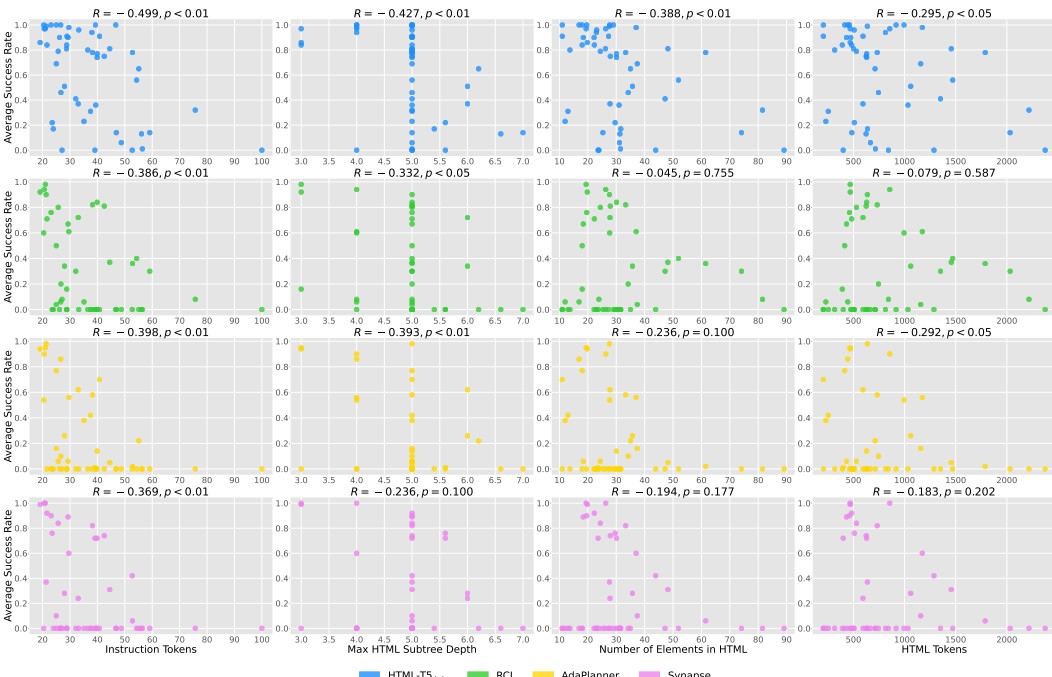

Figure 8: 2D-scatter plots between the success rate for each LMA (y-axis) and each statistic of compositional task (x-axis), such as the number of instruction tokens, max depth of HTML subtrees, the number of elements in HTML, and the number of HTML tokens. The results imply that while all the language model agents (HTML-T5++, RCI, AdaPlanner, Synapse) show negative correlations between the success rate and instruction tokens with statistical significance, the trends for other statistics may differ among the methods.

