# OpenReview forum: "Language Model Agents Suffer from Compositional Decision Making"
_ICLR.cc/2024/Conference — Submitted to ICLR 2024_

### Official Review · Reviewer_qv9D · 2023-10-21

**Soundness:** 3 good
**Presentation:** 3 good
**Contribution:** 3 good
**Rating:** 6
**Confidence:** 4

**Summary:**

The authors study the incompetence of LLMs in dealing with compositional decision making tasks, by proposing CompWoB, a new benchmark with 50 new compositional web automation tasks, training
HTML-T5++, a new model, with balanced data distribution across tasks, and empirically comparing with existing methods, including RCI, AdaPlanner, and Synapse.

**Strengths:**

originality

The authors study the incompetence of LLMs in dealing with compositional decision making tasks with a new benchmark and a new model with relatively comprehensive empirical study. It is novel.

quality

The paper is basically technically sound.

clarity

The paper is basically well-organized and clearly written.

significance

Language model-based agent becomes a buzz word, without carefully studying the capability of the foundational language models. The authors study the incompetence of LLMs in dealing with compositional decision making tasks. The community should carefully think how to make progress in language model-based agent, e.g., as recommended by the authors in the Discussion section, improving generalizable prompting methods, agent-specialized large language models, and parsing complex instructions to executable plan.

**Weaknesses:**

See questions below.

**Questions:**

1.
HTML-T5++ is an important contribution, which deserves a separate section, with more details of fine-tuning HTML-T5-XL, besides balancing data distribution.

2.
Can synthetic composing of web tasks represent realist ones? Are there ways to generate realist web tasks?

3.
Should web tasks be sequential decision making problems? That is, should there be dependencies between sub-web-tasks? Or simple composition of sub-tasks? How to achieve such dependancy?

If there is no dependancy among sub-tasks, why LLMs do not perform well on compositional tasks, which may be treated as multiple separated tasks? How to measure such dependancy?

4.
LLMs do not perform well at reverse-order instructions? Why? LLMs are widely regarded as being very competent with NLP tasks.

5.
"Figure 5 visualizes the correlation between the success rate averaged across WebGUM, HTML-T5, RCI, AdaPlanner, and Synapse (y-axis) and each statistic of compositional tasks (x-axis)"

Is such average success rate a good way?
Average may hide something.
Should we study each method individually, or the one with the best performance?

6. Some minor issues below

2 RELATED WORKS
Web Automation
"Although prior works have solved the problems with imitation learning and reinforcement learning ..."

6 RESULTS
“Otherwise mentioned, we adopt gpt-3.5-turbo as a backbone LLM.”
something wrong. how about "We adopt gpt-3.5-turbo as a backbone LLM, unless mentioned otherwise."

Figure 2
"and the dark color does in CompWoB"
Something wrong. How about "and the dark color for CompWoB"

Figure 3
Redundant info from Figure 2
"The light color represents the performance in CompWoB"
And the colors are different

---

> ### Author Response · Authors · 2023-11-20
> **Author Response to Reviewer eCh6**
>
> We thank the reviewer for the thoughtful review and comments. Please let us know any remaining questions or concerns if you have.
>
> **> Questions 1**
>
> Thank you for your suggestion. We separated the details of finetuned language model agents, including data-rebalancing, as an independent section in the revised paper (Section 4.5).
>
> **> Questions 2**
>
> While tasks are visually simplified, we believe CombWoB reflects a broader spectrum of functionalities in real-world websites without sacrificing controllability. For example, the task in Figure 1 sketches the structure of a login form with agreement checkboxes and a pop-up window, like Internet banking. We also designed tasks with page transitions, such as from the login form to the email browser. Some of the recent works utilized unsupervised auto-curricula to adaptively design new tasks from a set of primitive base tasks using a teacher/student paradigm [1, 2]. While they can be useful for designing realistic websites, they still suffer from a limited set of base tasks and training instability in a multi-agent system.
>
>
> **> Questions 3**
>
> Real-world websites have dependencies between sub-tasks. For example, Gmail requires successful login to proceed to writing an email or reading social media posts of a specific user requires navigating to the personal page of that user. In CombWoB, we implemented these kinds of dependencies in the reward function using logical AND operations. For example, while we allowed agents to continue to “write an email” subtask without successful login, the agent would get zero rewards even if the “write an email” subtask is successful. We inform agents of these dependencies using connectors in instructions such as “solve Y after X” or “solve X then Y”. Our analysis in Section 6.4 also suggests that, in addition to task compositionality, long instructions and deep HTML sub-tree can lead to challenging tasks.
>
>
> **> Questions 4**
>
> The failure analysis in Table 3  has shown that language model agents fail to parse the instructions into the correct order. This can be because base-task demonstrations in the prompt just have "left-to-right" instructions from base MiniWoB tasks; strongly conditioning the LLM agents to process instructions in a linear order while compositionality could imply a non-linear processing of instructions. This mismatch between the prompt ("left-to-right") and inference tasks (mixing "left-to-right" and "right-to-left") could cause the performance drops.
>
>
> **> Questions 5**
>
> Following the recent work in deep reinforcement learning literature [3], we measured average performance as a proxy of oracle task solvability. If many kinds of agents perform poorly, such tasks are regarded as challenging. This can shed light on "task" or "environment", rather than "language model agents" or "prompting methods", while such an analysis has been overlooked so far. Additionally, we agree with the reviewer that a single statistic, such as average over the methods, might not reflect the nuances for each method. While the trend with average might be the same, distributional characteristics for each language model agent could be different. We extended our analysis by reporting the individual performances of each agent in Appendix J (Figure 8). Our results still indicate that while all the language model agents (HTML-T5++, RCI, AdaPlanner, Synapse) often show negative correlations between the success rate and instruction tokens or max subtree depth with statistical significance, the trends for other statistics may differ among the methods.
>
> **> Questions 6**
>
> We appreciate your pointing out the grammatical errors in our manuscript. We fixed these errors based on your suggestions in the revised paper.
>
> ```
> [1] Gur et al., (2022) Environment Generation for Zero-Shot Compositional Reinforcement Learning (https://arxiv.org/abs/2201.08896)
>
> [2] Sohn et al., (2022) Fast Inference and Transfer of Compositional Task Structures for Few-shot Task Generalization (https://arxiv.org/abs/2205.12648)
>
> [3] Furuta et al., (2021) Policy Information Capacity: Information-Theoretic Measure for Task Complexity in Deep Reinforcement Learning (https://arxiv.org/abs/2103.12726).
> ```

---

> > ### Comment · Reviewer_qv9D · 2023-11-23
> >
> > Thanks for the reviews and authors' response. I will keep my score.

---

### Official Review · Reviewer_MfTP · 2023-10-30

**Soundness:** 3 good
**Presentation:** 3 good
**Contribution:** 2 fair
**Rating:** 6
**Confidence:** 3

**Summary:**

This paper proposes a new benchmark, called CompWoB – 50 new **compositional** web automation tasks reflecting more realistic assumptions. The authors then evaluate different LLMs to show that LLM-based agents suffer from compositional decision making. Detailed observations include: 1) while prompted gpt-3.5-turbo or gpt-4 achieve 94.0% average success rate on base tasks, their performance degrades to 24.9% success rate on compositional tasks; 2) transferred LLM-based agents (finetuned only on base tasks) show less generalization gap, dropping from 85.4% to 54.8%; 3) balancing data distribution across tasks, a finetuned model, HTML-T5++, surpasses human-level performance (95.2%) on MiniWoB, and achieves the best zero-shot performance on CompWoB (61.0%).

-----
after rebuttal, I increased the score to weak accept.

**Strengths:**

1. A noval and original study about the compositional web automation task is proposed and many insights are provided.

2. Propose a data distribution balancing method across tasks and finetune a new model to surpass human-level performance on MiniWoB.

3. Clear writting. The reviewer can follow most of this paper easily.

**Weaknesses:**

1. The reviewer did not get why Section 4 is needed (with such a large space), since most of the introductions are baseline methods.  Also, I did not know why RCI/AdaPlanner/Synapse are used for baselines.

2. Only test on 50 compositional web automation tasks. Are the methods and evaluations/insights generalizable to other tasks?

3. A lot of details are shown in the appendix (e.g., task difficulty estimation and data balancing method).

**Questions:**

1. why RCI/AdaPlanner/Synapse are used for baselines?

2. Are the methods and evaluations/insights generalizable to other tasks?

---

> ### Author Response · Authors · 2023-11-20
> **Author Response to Reviewer MfTP**
>
> We appreciate your careful reading and detailed discussion of our paper. We address your concerns below and please let us know if there are remaining questions or unclear points.
>
> **> Weaknesses 1 & Questions 1**
>
> We selected baseline LLM agents in our work based on their superior performance and novelty in using LLMs for web navigation problems. RCI [1] is the first to use prompting in a self-refinement loop, outperforming SL/RL agents on MiniWoB benchmark that requires millions of demonstrations to work. AdaPlanner [2] and Synapse [3] were the follow-up works outperforming RCI via code generation from environmental feedback or via well-designed decomposed prompts with retrieval.
>
> Since we extensively studied these three reference models, we wanted to provide a detailed and to-the-point summary of these models so that it is easy to reason about their performance, shortcomings, and practicality. To improve readability, we shortened Section 4 by moving some of the content to the Appendix.
>
>
>
> **> Weaknesses 2 & Questions 2**
>
> MiniWoB has around 104 base tasks; all the two-way and three-way combinations of these tasks would give 185,460 compositional tasks. It is infeasible to manually curate all the two-way/three-way combinations. Unfortunately, it is also nontrivial to automate this process due to the locality of the environment implementations in MiniWoB. Each environment is mostly self-contained, making it nontrivial to modularize the whole benchmark/codebase so that every combination can be automatically generated.
> We decided to outline a set of design principles – solvability, feasibility, reality, etc. which we follow to manually curate 50 compositional tasks and 50 reverse-order instruction tasks that we believe cover a wide variety of difficulty, and compositionality. We believe these guiding principles would be applicable to other compositional generalization problems such as robotic navigation. We also believe that based on these design principles, our compositional benchmark can easily be extended in the future to study even more challenging and compositional web navigation problems.
>
>
> We'd also like to highlight that, because previous works were tested around 50 - 60 MiniWoB tasks [1,2,3,4,5], 50 compositional tasks is a decent number of tasks to evaluate the agents. Our addition of 50 new compositional tasks doubles the number of tasks for the community to study.
>
>
>
>
> **> Weaknesses 3**
>
> Due to limited space, we had to carefully structure our paper. We wanted to make the main part to be as self-contained as possible while also giving more details in the appendix for clarification and reproducibility. We made some changes to include some of the details of data balancing and task difficulty estimation as a new section in the revised paper (Section 4.5). We explain how difficulty scores are estimated (Appendix D), or the ratio of tasks in the rebalanced dataset (Appendix E) in the appendix. Please let us know if you have any other concerns.
>
>
>
> ```
> [1] Kim et al., (2023) Language Models can Solve Computer Tasks (https://arxiv.org/abs/2303.17491)
>
> [2] Sun et al., (2023) AdaPlanner: Adaptive Planning from Feedback with Language Models (https://arxiv.org/abs/2305.16653)
>
> [3] Zheng et al., (2023) Synapse: Trajectory-as-Exemplar Prompting with Memory for Computer Control (https://arxiv.org/abs/2306.07863)
>
> [4] Gur et al., (2022) Understanding HTML with Large Language Models (https://arxiv.org/abs/2210.03945)
>
> [5] Furuta et al., (2023) Multimodal Web Navigation with Instruction-Finetuned Foundation Models (https://arxiv.org/abs/2305.11854)
> ```

---

> > ### Comment · Reviewer_MfTP · 2023-11-22
> > **The response addresses most of my concerns**
> >
> > I thank the authors for detailed responses. It clasifies the reasons why the baselines and evaluation tasks are used. It also add more details in the appendix. From the reviewer perspective, these responses are reasonable and address most of my concerns, and I would like to increase my rating score.

---

### Official Review · Reviewer_G9qb · 2023-10-31

**Soundness:** 4 excellent
**Presentation:** 4 excellent
**Contribution:** 3 good
**Rating:** 8
**Confidence:** 4

**Summary:**

This paper looks at the ability of LMAs to solve compositional web-tasks. A new dataset is introduced based on the existing Mini-WoB. Models are prompted with base tasks and then asked to solve tasks that are composed of different base tasks. Experiments show that performance drops across both LMAs and fine-tuned models.

**Strengths:**

- The topic of compositionally in web tasks  is extremely important given how many papers have been released in the past year showing that GPT can be used for web tasks.
- A new dataset is introduced which can show how well LMAs actually do given a combination of tasks without any prompting. One strong aspect of the benchmark is that consists of individual tasks that LLMs already know how to solve so it is clear that the difficulty is in combining tasks.
- The paper is well written and has a thorough analysis about the different results. In particular, section 6.4 gives insight into what makes tasks more difficult, something not usually addressed.

**Weaknesses:**

- For LMAs, there is no discussion on how the prompt could be modified for combining tasks. For example, if a prompt shows how to perform a joint task, is the performance any better?

**Questions:**

None (see above)

---

> ### Author Response · Authors · 2023-11-20
> **Author Response to Reviewer G9qb**
>
> We appreciate the careful reading and thoughtful comments. We address your concerns below, and please let us know if there are remaining questions or unclear points.
>
>
> **> Weaknesses 1**
>
> We provided additional results with oracle demonstrations for compositional tasks in Appendix I (Figure 7). RCI with GPT-4 and oracle exemplars achieves 82.9% success rate averaged on 20 two-way tasks, which is the best among the methods (e.g. HTML-T5++: 73.9%, RCI with gpt-3.5-turbo: 46.9%, RCI with gpt-4: 71.5). This indicates that if a prompt includes how to perform on compositional tasks, the performance gets better. However, please keep in mind that providing exemplars for every compositional problem is infeasible given the huge space of problems and the cost of collecting examples for every one of them.

---

> > ### Comment · Reviewer_G9qb · 2023-11-22
> >
> > Thank you for your comments. Based on comments and other reviews, my initial score of 8 does not change.

---

### Official Review · Reviewer_PK52 · 2023-11-01

**Soundness:** 3 good
**Presentation:** 4 excellent
**Contribution:** 2 fair
**Rating:** 6
**Confidence:** 3

**Summary:**

The authors propose a web automation agent model and test it on a proposed "compositional" benchmark. They show that standard general-purpose language model agents have their performance deteriorate more on their proposed benchmark than models fine-tuned on similar tasks.

**Strengths:**

This is solid research that asks and answers a somewhat important question. It is thorough, with a reasonable set of agent techniques and a reasonable methodology for extending MiniWoB.

**Weaknesses:**

The contribution is relatively minor (which, in my view, is fine - obviously, not every ICLR paper needs to be revolutionary). This is especially true because "compositionality" is inherently somewhat arbitrary: the tasks in MiniWoB are arguably already compositional since they require a series of steps performed in the right order. By the same reasoning, arguably, all language model hierarchical/long-range planning papers, not to mention several multimodal language model approaches designed to reason over images, are performing compositional tasks. I'd also point out that there are specific strategies that have been proposed specifically for compositional action (e.g., Parsel from Zelikman et al. 2022, which uses LMs to propose a high-level plan in language and implements each subpart independently).

Some nitpicks: The title makes it sound like the model itself is harmed, but that doesn't really make sense. And, in conjunction with the earlier point about MiniWoB also being somewhat compositional, the title isn't necessarily backed up by the experiments. I think this could be easily partially fixed by simply adding "for web automation" to the title after LMA, and web automation is probably relevant enough that with this narrower scope, it's still fine. I expect I would lower my score if the authors don't commit to making this or some other disambiguating change.

**Questions:**

See limitations

---

> ### Author Response · Authors · 2023-11-20
> **Author Response to Reviewer PK52**
>
> We thank the reviewer for the careful reading and constructive feedback. We address your concerns below and please let us know if you have further questions.
>
> **> Weaknesses 2**
>
> Thank you for your suggestions. We decided to change the title to `Language Model Agents Suffer from Compositional Generalization in Web Automation` in the final version. Since we intentionally distinguish the terminology of "language model agents" (prompting & pipeline) and "large language models" (model itself), we do not intend to argue "the model itself is harmed" in the title; in fact, we have compared different agent-prompting methods (RCI, AdaPlanner, Synapse) on the same LLMs (gpt-3.5-turbo/gpt-4). Please let us know if you have further concerns about the title.
>
> **> Weaknesses 1**
>
> We agree with the reviewer that many of the real-world tasks have inherent compositionality to some degree, including MiniWoB tasks. However, these tasks are not explicitly designed for compositionality, making it difficult to systematically investigate the generalization gap. For example, existing language model agents, that we also study in our paper, already achieve human-level performance on MiniWoB – more than 90% performance on almost all the tasks in MiniWoB. However, when these tasks are combined, their performance drops significantly – allowing us to analyze this gap using the difficulty of the base tasks themselves as well as the difficulty of the task compositions. Given that these agents already solve base tasks, we found it surprising that even the most capable GPT-4-based agents still struggle with task compositions while a fine-tuned T5-based model transfers better. We also added Parsel to related work (Section 2) in the revised paper.

---

> > ### Comment · Reviewer_PK52 · 2023-11-22
> >
> > Thank you for your response! I will keep my score as is.

---

### Author Response · Authors · 2023-11-20
**Summary of Revision in Author Response**

We would like to appreciate the thoughtful comments from all the reviewers. We revised the manuscript based on your constructive feedback and suggestions (*highlighted in purple*).

First, considering the suggestion from reviewer PK52, we'd like to change the title in the revised paper.

- (old) Language Model Agents Suffer from Compositional Decision Making
- (new) **Language Model Agents Suffer from Compositional Generalization in Web Automation**

The other key changes are summarized below:

- Add an independent section about task difficulty estimation and data-rebalancing for finetuned language model agents (Section 4.5).
- Reduce the description of Synapse (Section 4.3).
- Include additional related work (Section 2).
- Add additional results of prompted language model agents with oracle demonstrations (Appendix I).
- Add additional task complexity analysis for each language model agent (Appendix J)


We hope our revision and response to each reviewer address your concerns. Let us know any remaining questions or concerns if you have.

---

### Meta-Review · Area_Chair_AuKn · 2023-12-19

**Metareview:**

A new compositional benchmark for web automation.

Reviewers found the manuscript to be sound. The benchmark is reasonable and is compositional. Compositional reasoning has been a long-standing problem in machine learning, and modern models can still struggle with it. The benchmark demonstrates this through the low performance that even the best models have when information is held out.

That being said, as the reviewers point out, this is a minor contribution. There is nothing that makes this benchmark stand out compared to other compositionality benchmarks, other than the use of web automation. While this is a domain of great interest to some, that on its own is not dispositive. Scientifically, the reviewers are correct, this is a variation of prior work. It is unclear what the wider ICLR community can take away from this benchmark compared to other work.

**Justification For Why Not Higher Score:**

There is not much for an ICLR audience to learn from this manuscript, there are numerous other compositional benchmarks available now.

**Justification For Why Not Lower Score:**

N/A

---

### Decision · Program_Chairs · 2024-01-16

Reject